# Genetic variation in *ALDH4A1* is associated with muscle health over the lifespan and across species

**Osvaldo Villa**[1†], **Nicole L Stuhr**[1,2†], **Chia-an Yen**[1,2†], **Eileen M Crimmins**[1], **Thalida Em Arpawong**[1], **Sean P Curran**[1,2,3]*

[1]Leonard Davis School of Gerontology, University of Southern California, Los Angeles, United States; [2]Dornsife College of Letters, Arts, and Science, Department of Molecular and Computational Biology, University of Southern California, Los Angeles, United States; [3]Norris Comprehensive Cancer Center, University of Southern California, Los Angeles, United States

**Abstract** The influence of genetic variation on the aging process, including the incidence and severity of age-related diseases, is complex. Here, we define the evolutionarily conserved mitochondrial enzyme ALH-6/ALDH4A1 as a predictive biomarker for age-related changes in muscle health by combining *Caenorhabditis elegans* genetics and a gene-wide association scanning (GeneWAS) from older human participants of the US Health and Retirement Study (HRS). In a screen for mutations that activate oxidative stress responses, specifically in the muscle of *C. elegans*, we identified 96 independent genetic mutants harboring loss-of-function alleles of *alh-6*, exclusively. Each of these genetic mutations mapped to the ALH-6 polypeptide and led to the age-dependent loss of muscle health. Intriguingly, genetic variants in *ALDH4A1* show associations with age-related muscle-related function in humans. Taken together, our work uncovers mitochondrial *alh-6/ALDH4A1* as a critical component to impact normal muscle aging across species and a predictive biomarker for muscle health over the lifespan.

*For correspondence:
spcurran@usc.edu

†These authors contributed equally to this work

**Competing interest:** The authors declare that no competing interests exist.

## Editor's evaluation

Age-associated muscle decline is a pervasive aspect of human aging biology. Here the authors provide substantial genetic evidence implicating *C. elegans* mitochondrial enzyme ALH-6 (P5C dehydrogenase) in muscle oxidative stress and in the maintenance of muscle functionality late into life. The authors also analyzed databases on human aging to identify a linkage of human ALH-6 homolog ALDH4A1 and indicators of human muscle function in aging, which suggests conserved function of ALH-6/ALDH4A1 in aging and the potential for use of ALDH4A1 genetic data as a predictor for old age muscle health.

## Introduction

Sarcopenia is defined as the age-related degeneration of skeletal muscle mass and is characterized by a progressive decline in strength and performance (*Santilli et al., 2014*). This syndrome is prevalent in older adults and has been estimated by large scale studies to afflict 5–13% of people aged 60–70 years and expands to 50% of those aged 80 and above (*von Haehling et al., 2010*). Loss of muscle function is associated with a decline in quality of life and higher mortality and morbidity rates due to increased chance of falls and fractures (*Tsekoura et al., 2017*; *Ahmadpanah et al., 2015* ). Sarcopenia is linked to risk factors, such as a sedentary lifestyle, lack of exercise, and a diet deficient in protein

**eLife digest** Ageing is inevitable, but what makes one person 'age well' and another decline more quickly remains largely unknown. While many aspects of ageing are clearly linked to genetics, the specific genes involved often remain unidentified.

Sarcopenia is an age-related condition affecting the muscles. It involves a gradual loss of muscle mass that becomes faster with age, and is associated with loss of mobility, decreased quality of life, and increased risk of death. Around half of all people aged 80 and over suffer from sarcopenia. Several lifestyle factors, especially poor diet and lack of exercise, are associated with the condition, but genetics is also involved: the condition accelerates more quickly in some people than others, and even fit, physically active individuals can be affected.

To study the genetics of conditions like sarcopenia, researchers often use animals like flies or worms, which have short generation times but share genetic similarities with humans. For example, the worm *Caenorhabditis elegans* has equivalents of several human muscle genes, including the gene *alh-6*. In worms, *alh-6* is important for maintaining energy supply to the muscles, and mutating it not only leads to muscle damage but also to premature ageing. Given this insight, Villa, Stuhr, Yen et al. wanted to determine if variation in the human version of *alh-6*, *ALDH4A1*, also contributes to individual differences in muscle ageing and decline in humans.

Evaluating variation in this gene required a large amount of genetic data from older adults. These were taken from a continuous study that follows >35,000 older adults. Importantly, the study collects not only information on gene sequences but also measures of muscle health and performance over time for each individual. Analysis of these genetic data revealed specific small variations in the DNA of *ALDH4A1*, all of which associated with reduced muscle health.

Follow-up experiments in worms used genetic engineering techniques to test how variation in the worm *alh-6* gene could influence age-related health. The resulting mutant worms developed muscle problems much earlier than their normal counterparts, supporting the role of *alh-6/ALDH4A1* in determining muscle health across the lifespan of both worms and humans.

These results have identified a key influencer of muscle health during ageing in worms, and emphasize the importance of validating effects of genetic variation among humans during this process. Villa, Stuhr, Yen et al. hope that this study will help researchers find more genetic 'markers' of muscle health, and ultimately allow us to predict an individual's risk of sarcopenia based on their genetic make-up.

and micronutrients (*Tsekoura et al., 2017*). However, several aspects of the molecular basis of the age-dependent decline in muscle health remain unknown.

Although age-related muscle function is clearly linked to frailty (*Dent et al., 2019*), previously, different etiologies of clinical weakness led to discrepancies in the definitions of sarcopenia (*Batsis et al., 2013*; *Cruz-Jentoft et al., 2010*). Furthermore, the identification of human genetic loci that influence age-related functions has traditionally been difficult to characterize due to the methodological difficulties in longitudinal assessments; the prevalence of sarcopenia for example begins in the fourth decade of life (*Sayer et al., 2008*). The US Health and Retirement Study (HRS) is a nationally representative survey of adults aged 50 years and older and has proven to be an invaluable dataset for investigating the normal aging processes (*Fisher and Ryan, 2018*; *Sonnega et al., 2014*; *Juster and Suzman, 1995*; *HRS Health and Retirement Study, 2021*). New skeletal muscle cutpoints for identifying elevated risk for physical disability in older adults (*Janssen et al., 2004*) have enabled cross-sectional analyses to identify cohorts of HRS participants with age-related decline in muscle function (e.g., grip strength basic activities of daily living [ADL] and instrumental ADL [IADL]) (*MacEwan et al., 2018*).

In *Caenorhabditis elegans*, mutation of the conserved proline catabolic gene *alh-6* (88% identity to *ALDH4A1* in humans) leads to premature aging and impaired muscle mitochondrial function (*Pang and Curran, 2014*). Proline catabolism functions in a two-step reaction, beginning with the conversion of proline to 1-pyrroline-5-carboxylate (P5C) which is catalyzed by proline dehydrogenase, PRDH-1; subsequently, P5C dehydrogenase, ALH-6, catalyzes the conversion of P5C to glutamate. *alh-6* expression is observed in body wall muscle, pharyngeal muscle, and neurons (*Pang and Curran, 2014*), and when *alh-6* is mutated, the activation of *gst-4p::gfp* oxidative stress reporter is predominantly

observed in the body wall muscle tissue and only in postreproductive adults (*Tang and Pang, 2016*). *alh-6* mutants have increased levels of P5C; the accumulation of this toxic metabolic intermediate leads to an increase in reactive oxygen species, including $H_2O_2$ (*Pang and Curran, 2014*), which then activates cytoprotective responses, impairs mitochondrial activity, and drives cellular dysfunction (*Pang and Curran, 2014*; *Pang et al., 2014*; *Yen et al., 2020*; *Yen and Curran, 2021*).

Several studies have linked disease states that drive morbidity and mortality with genomic variation through genome-wide association studies (*Timmers et al., 2019*; *Tam et al., 2019*; *Manolio et al., 2009*) and nonhuman models have been utilized to test how single genes can drive phenotypes that mimic the disease state in humans (*Ke et al., 2021*; *Song et al., 2020*; *Teumer et al., 2019*). However, biological testing of genetic association studies for the normal human aging process remains under-represented. The recent expansion of the HRS data to include genotyping of participants has enabled scans to test associations between normal aging phenotypes and variation across genes (*Liu et al., 2019*). In this study, we exploit the facile genetic tractability of *C. elegans* with the rich genetic and phenotypic data available in the HRS to reveal genetic variation in *alh-6/ALDH4A1* as a predictive indicator of muscle-related functionality in later life.

## Results and discussion

While the strong induction of oxidative stress reporter activity in the musculature was linked to mutation of the mitochondrial P5C dehydrogenase gene, and not observed in other genetic mutants (*Pang et al., 2014*; *Yen et al., 2020*; *Yen and Curran, 2021*; *Lo et al., 2017*; *Spatola et al., 2019*; *Lynn et al., 2015*; *Nhan et al., 2019*), the breadth of genetic mutations that could induce stress responses in muscle was unknown. In order to identify additional genetic components of this age-related muscle phenotype, we performed an ethyl methanesulfonate mutagenesis screen selecting for the same age-dependent activation of the *gst-4p::gfp* reporter in the musculature. We screened the progeny of ~4000 mutagenized F1 animals and isolated 96 mutant animals with age-dependent activation of the *gst-4p::gfp* reporter restricted to the body wall musculature, which phenocopies the *alh-6(lax105)* mutant (*Figure 1a*, *Figure 1—figure supplement 1*). To rule out additional loss-of-function alleles of *alh-6* we performed genetic complementation (cis–trans) testing with our established *alh-6(lax105)* allele; surprisingly, all 96 new mutations failed to complement and as such were all loss-of-function alleles of *alh-6*.

To catalog these mutations, we began sequencing the *alh-6* genomic locus in each of the mutants isolated. After sequencing approximately half of the mutants, we noted the repeated independent isolation of several distinct molecular lesions in *alh-6*: E78Stop (*lax903*, *lax918*, *lax930*), E447K (*lax916*, *lax920*, *lax929*, *lax934*), G527R (*lax914*, *lax932*, *lax933*, *lax947*), etc. (*Figure 1b*). The lack of diversity in genes uncovered and the independent isolation of identical alleles multiple times from this unbiased screen strongly suggest genetic saturation and specificity of this response to animals with defective mitochondrial proline catabolism. In addition, several mutations mapped to discreet regions of the linear ALH-6 polypeptide, including G152/K153, G199/E201/G202, and E418/G419, which may define critical domains in the folded protein. Imaging at day 3 of adulthood revealed that each mutant was phenotypically identical to *lax105* in the activation of the *gst-4p::gfp* stress reporter in the bodywall muscle (*Figure 1—figure supplement 1*), but with varying intensity (*Figure 1c*). We next mapped the location of each amino acid mutated in our panel of *alh-6* mutants on the structure of the ALH-6 protein (*Figure 1—figure supplement 2*; *Kelley et al., 2015*; *Ittisoponpisan et al., 2019*), which enabled a prediction model of the impact of each missense mutation (*Figure 1—source data 1*). Most missense mutations were predicted to maintain the overall structure (no structural damage), which suggests the associated phenotypes derive from a range of reduction of function mutations. Since the degree of mitochondrial dysfunction can influence both beneficial and detrimental physiological outcomes (*Wang and Hekimi, 2015*; *Shields et al., 2021*), this collection of mutants provides a model to understand the complex role mitochondria play in organismal health over the lifespan. Taken together, these data reveal that the age-dependent and muscle activation of the *gst-4p::gfp* is driven specifically by mutations in mitochondrial *alh-6*.

Based on the striking specificity of the muscle-restricted and age-dependent activation of the *gst-4p::gfp* stress reporter in *C. elegans* harboring mutations in *alh-6* (*Pang and Curran, 2014*; *Yen et al., 2020*), combined with the high degree of conservation in mitochondrial metabolism pathways across metazoans (*Pang et al., 2014*), we reasoned that *ALDH4A1* genetic variants would associate with

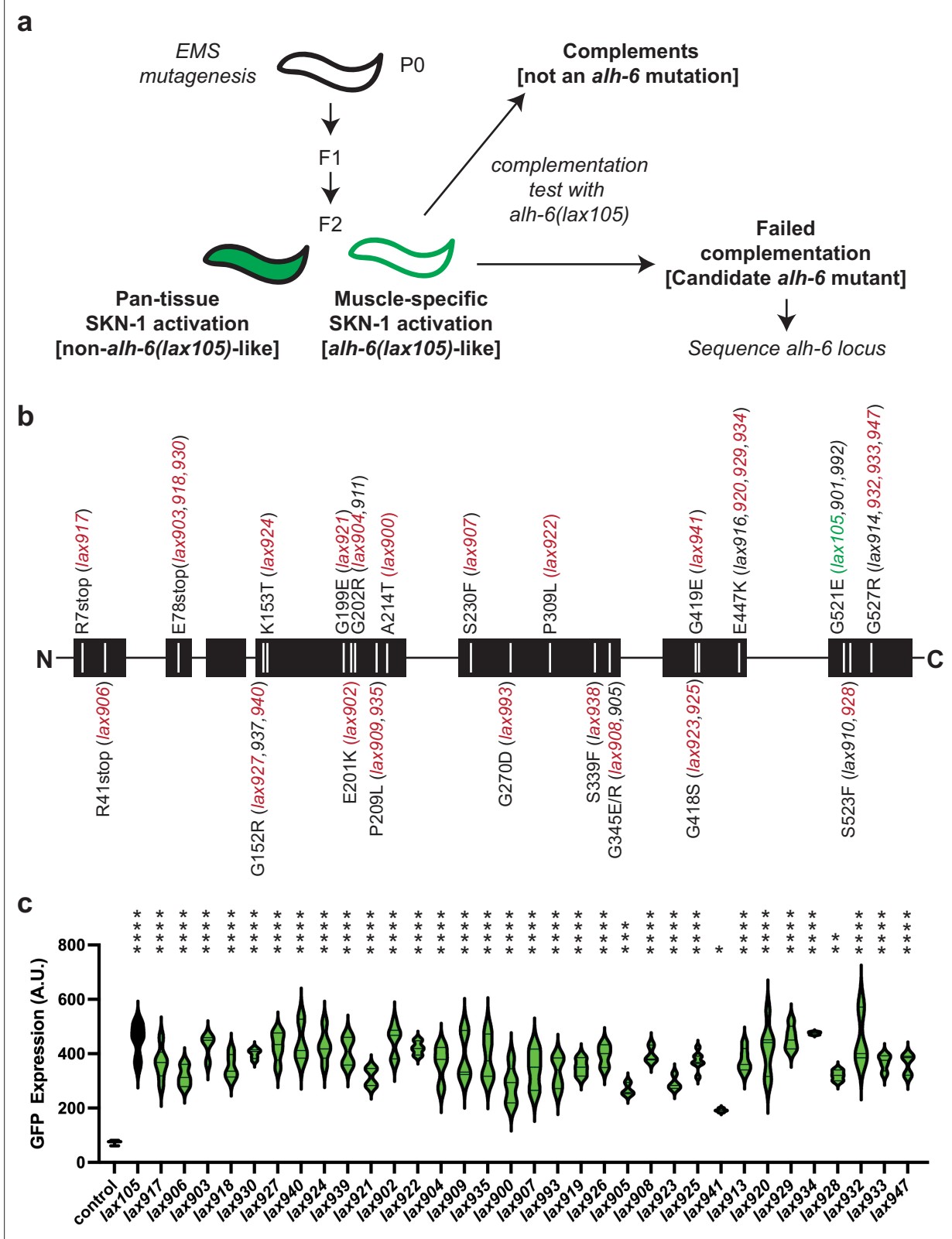

**Figure 1.** Mutation of *alh-6* uniquely activates age-dependent and activation of the *gst-4p::gfp* oxidative stress reporter in muscle. (**a**) Schematic representation of genetic screen for mutants that phenocopy *alh-6(lax105)*. (**b**) Schematic representation of the ALH-6 protein with the molecular identity of mutants isolated and sequenced annotated. Alleles that were selected for additional functional tests of muscle function (*Figure 4*) are highlighted in red and the location of the canonical *alh-6(lax105)* allele is highlighted in green. These alleles represent all the sequenced mutations in *alh-6* that were

*Figure 1 continued on next page*

*Figure 1 continued*

isolated from the ethyl methanesulfonate (EMS) screen. (**c**) Quantification of stress reporter activation in the muscle in the new *alh-6* mutant alleles, as measured by the intensity of GFP fluorescence from the oxidative stress reporter *gst-4p::gfp* (see *Figure 1—figure supplement 1* for representative images). *t*-Test relative to *gst-4p::gfp* reporter animals (control); *p < 0.05; **p < 0.01; ***p < 0.001; ****p < 0.0001.

The online version of this article includes the following source data and figure supplement(s) for figure 1:

**Source data 1.** Structure-function predictions of ALH-6 mutant proteins from computational modeling.

**Figure supplement 1.** Novel alleles of *alh-6* induce muscle-specific activation of the *gst-4p::gfp* stress reporter reporter.

**Figure supplement 2.** Location of amino acid substitutions in *alh-6* mutants.

phenotypes indexing normative, longitudinal changes in human aging-related functionality, specifically those that involve usage of different muscle groups. To test this hypothesis, we performed gene-wide association scanning (GeneWAS) adjusting for relevant covariates and indicators of population stratification in the US HRS; a nationally representative longitudinal study of >36,000 adults over age 50 in the United States (*Sonnega et al., 2014*; *Weir, 2013*). HRS collects biological and genetic samples on subsets of participants and assesses physical and psychosocial measures of all study participants in older adulthood, including multiple measures of muscle-related functionality (*Figure 2—source data 1*). The human phenotypes, represented in the HRS index normative changes in aging-related physiological ability. There were 70 single nucleotide polymorphism (SNP) markers within the *ALDH4A1* region that are on the Illumina Omni array representing 273 human SNPs in the gene. While measures like grip strength are more commonly used to assess muscle health, our inclusion of another phenotype represents changes in complex physiological process that are influenced by the musculature and also other systems (e.g., metrics of walking speed can also be influenced by neurological factors). As such, future work to assess the role of neuronal *alh-6/ALDH4A1* will be important. Nevertheless, the observed decline in muscle-related measures with age is relevant. Overall, two associations between variants within *ALDH4A1* and two phenotypes were detected and surpassed the respective empirical p value thresholds, determined by permutation testing (*Tables 1 and 2*). These demonstrate a pattern of association between *ALDH4A1* variation and two independent phenotypes (*Table 1*). Because each of the SNPs within the *ALDH4A1* region represents a tag, or marker SNP for human variation within the locus, this GeneWAS was unable to directly identify a causal variant; however, the indexing of variations within the same gene suggests conserved associations within this aging human cohort.

With this study design, we did not intend to find a single genetic variant that would explain functionality of a specific muscle group. Specifically, we chose to include common measures of physical functioning that index aging-related decline. We calculated phenotypes for decline in functionality over time because they are more robust for testing genetic associations, represent normative aging processes in human samples compared to single-time point assessments, and index broader human functionality. (*Figure 2—source data 1*). These results indicate that variants within the *ALDH4A1* locus affect an individual's performance on basic ambulatory movements such as speed of walking short distances or ability to exert hand grip strength.

rs77608580 was significantly associated with change in gait speed over time (*Figure 2a*). Specifically, with each additional A allele, there was an average increase in gait speed of 0.052 m per second per year compared to other same aged individuals without the allele (p value = 0.0025, surpassing the empirical p value threshold of 0.006). This was assessed among *N* = 3319 older individuals with a mean age of 73.0 years (standard deviation [SD] = 5.9) and mean gait speed of 0.80 m per second (SD = 0.25), or 2.6 ft/s (*Figure 3a*).

**Table 1.** Top SNPs associated with specific phenotypes.

| Phenotype | SNP name | Location | Ref. allele | Minor allele | Freq.* | Scan N† | Effect size‡ | p value |
|---|---|---|---|---|---|---|---|---|
| Grip strength decline | rs28665699 | 19200185 | A | A | 0.014 | 5228 | −0.045 | 9.1E−04 |
| Gait speed decline | rs77608580 | 19196968 | A | G | 0.017 | 3319 | 0.052 | 2.5E−03 |

*Freq = frequency of minor allele as reported by 1000 genomes.
†N = sample size of scan for the phenotype and SNP.
‡Effect sizes provided are standardized regression coefficients.

**Table 2.** SNPs remaining after filtering for minor allele frequency and pruning based on linkage disequilibrium.

| SNP | Location | Reference allele | Minor allele frequency |
|-----|----------|------------------|------------------------|
| rs28652778 | 19194995 | A | 0.20 |
| rs28405179 | 19195143 | A | 0.03 |
| rs111289603 | 19195492 | G | 0.03 |
| kgp2515954 | 19195951 | A | 0.02 |
| rs77608580 | 19196968 | A | 0.04 |
| rs9699485 | 19197237 | G | 0.02 |
| rs3935824 | 19197849 | G | 0.18 |
| rs28665699 | 19200185 | A | 0.03 |
| rs28493067 | 19203333 | A | 0.35 |
| rs6426814 | 19204173 | A | 0.19 |
| rs35285457 | 19205258 | A | 0.14 |
| rs7365978 | 19206020 | A | 0.21 |
| rs28508407 | 19210018 | A | 0.27 |
| rs113232075 | 19211163 | G | 0.02 |
| rs9426718 | 19213022 | A | 0.02 |
| rs4911985 | 19215440 | G | 0.22 |
| rs28582076 | 19217295 | G | 0.02 |
| rs11484743 | 19219987 | C | 0.02 |
| rs17492518 | 19221621 | A | 0.04 |
| rs4912044 | 19230263 | A | 0.18 |
| rs79251057 | 19231130 | A | 0.04 |

Measures of muscle health, such as grip strength, are effective biomarkers of overall health in older populations (**Carson, 2018**; **McGrath et al., 2020**). rs28665699 was significantly associated with an increase in grip strength over time (**Figure 2b**); with each additional A allele, there was an average increase in grip strength by 0.045 kg weight per year while holding all other characteristics constant (age, sex, and use of the dominant hand for gripping; p value = 0.0009, surpassing the empirical p value threshold of 0.0019). This was assessed among $N$ = 5228 older individuals with a mean age of 68.9 years (SD = 10.4), mean grip strength of 30.21 kg (SD = 11.1), and average level of decline in grip strength at 2.31 kg per year (SD = 5.37). If calculated as change over a 10-year period, those with one or two effect alleles would have stronger grip by 0.5 and 1.0 kg compared to those without an effect allele, respectively. The allele therefore is associated with a slower rate of decline in grip strength over a decade of age (**Figure 3b**).

These effects are examples where variation in the gene contributing to phenotypes that represent different age-related change in functionality; overall we find there are small effects associated with each phenotype, but there are possible pleiotropic effects, and environmental or behavioral factors contributing. It is not known if any one of the identified *ALDH4A1* SNPs is a causal variant or if they mark a different variant within the *ALDH4A1* gene that was not represented on the HRS array. Regardless, these results collectively support a true association between *ALDH4A1* and age-related physical function.

We tested replication of the top two SNPs from the GeneWAS across ethnic subsamples in the HRS by calculating a common effect size across the samples. We did this by completing a fixed effects and random effects meta-analysis using PLINK software (**Rentería et al., 2013**; **Purcell et al., 2007**; **Table 3**). For one SNP, the minor allele frequency in the African ancestry sample was below 1% and thus the subsample was excluded from the meta-analysis. The Cochrane's Q statistic (Q), as an indicator of variance across sample effect sizes, and the heterogeneity index (I), which quantifies dispersion across samples indicate random effects analysis fit the data better for gait speed decline, thus we focus on results from random effects to account for differences in effect sizes by sample (e.g., the I index indicates 64.95% of the observed variance between samples is due to differences in effect sizes between samples). Given these results, the common effect size calculated for grip strength decline still suggest significance of these associations with SNPs in the gene, whereas the effect for gait speed decline remained for the European ancestry cohort only and not across subsamples. Genetic data obtained from similarly large international cohorts studies (e.g., English Longitudinal Study of Ageing [ELSA; https://www.elsa-project.ac.uk]; Irish Longitudinal Study on Ageing [TILDA; https://tilda.tcd.ie/]; cohorts in the Survey of Health, Ageing and Retirement in Europe [SHARE; https://g2aging.org/overviews?study=share-aut], or Northern Ireland Cohort for the Longitudinal Study of Ageing [NICOLA; https://www.qub.ac.uk/sites/NICOLA/AboutNICOLA/]; and others who are aged 50 and older will enable additional replication and additional cross comparisons).

To test how genetic variation in P5C dehydrogenase can influence age-related muscle function, we returned to our collection of *C. elegans* strains harboring mutations in *alh-6*. We measured

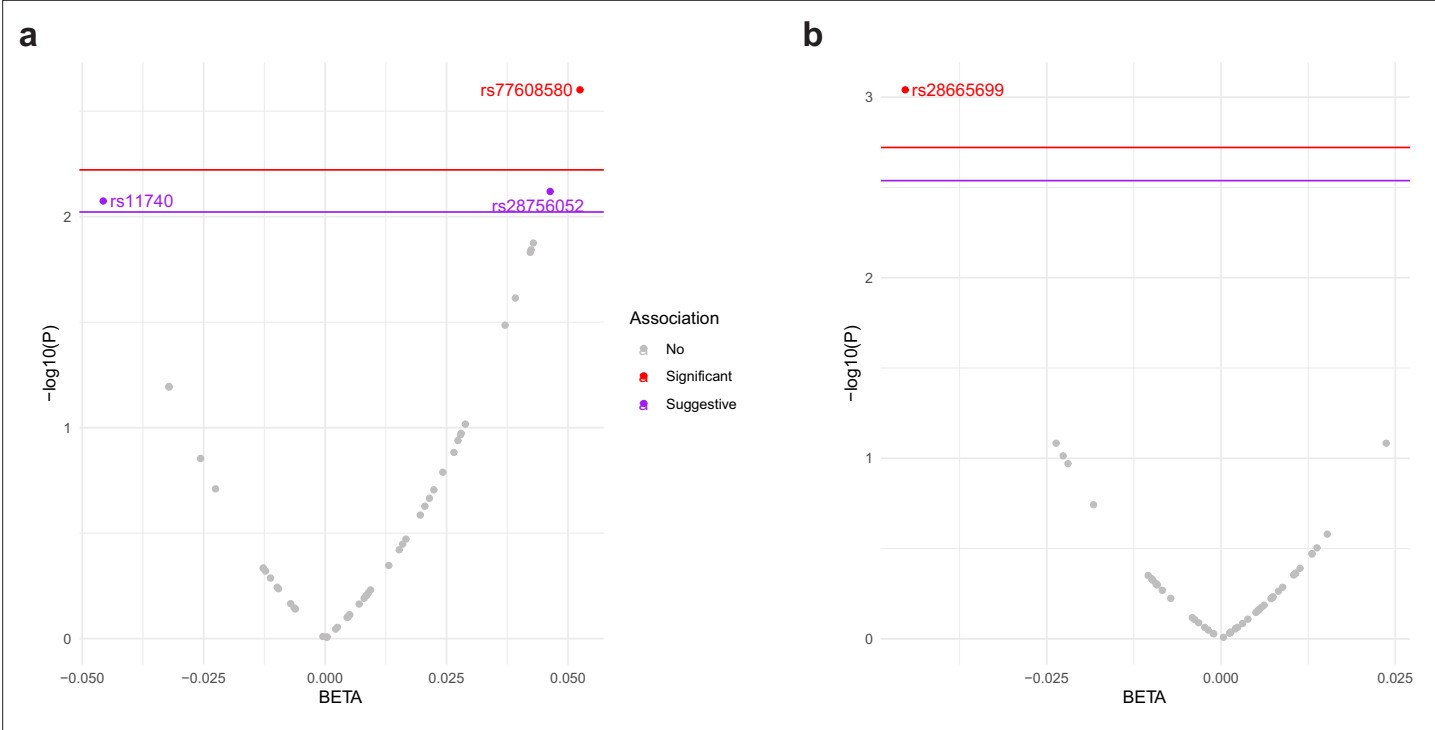

**Figure 2.** *ALDH4A1* variants associate with human age-related phenotypes for change in muscle function. Plot of association between variants in the *ALDH4A1* gene and normative aging-related muscle decline in (**a**) gait speed and (**b**) grip strength in the US Health and Retirement Study (HRS). The *x*-axis shows the beta estimate for the effect of each SNP, represented by a dot, on the phenotype. The *y*-axis shows the log of the p value for the association between the SNP and the phenotype. SNPs that surpassed the empirical p value threshold, shown as a red line, for decline in gait speed (empirical p value = 0.006) and grip strength (empirical p value = 0.0019) are depicted as red dots. SNPs that surpassed a suggestive threshold (p value = 0.009 for gait speed) are depicted as purple dots.

The online version of this article includes the following source data for figure 2:

**Source data 1.** Details for phenotypes calculated from the US Health and Retirement Study.

individual animal movement speed as a function of muscle health with age (*Roussel et al., 2014*). Only animals harboring mutations of ALH-6 at position G152R(*lax940*), K153T(*lax924*), S523F(*lax928*), and G527R(*lax933*) resulted in a significant loss of movement speed at larval stage 4; just prior to adulthood (*Figure 4a*). However, with the exception of S230F(*lax907*) and Y427N(*lax918*), all mutants tested displayed a significant reduction in movement speed at day 3 of adulthood (*Figure 4b*). As a secondary measure of muscle function in our panel of *alh-6* mutants, we measured changes in swimming performance (*Figure 4—figure supplement 1*), which has documented effects on animal health and longevity (*Laranjeiro et al., 2019*). It is established that swimming is a more energetically demanding activity than crawling on a plate (*Laranjeiro et al., 2017*). Intriguingly, the effect of the canonical *alh-6(lax105)* mutation on swimming was less pronounced than that observed for crawling speed and our panel of *alh-6* mutants displayed differences in developmental and adult swimming performance. Taken together these data support the age-specific acceleration of muscle decline in mitochondrial proline catabolism mutants, which is conserved from nematodes to humans.

The traits analyzed in the HRS came from a population-based study and were not assessed to allow us to identify physiological degeneration in specific muscles, rather to index and track overall age-related decline in functionality over time. It is widely accepted that genetic variation underlying these aging-related traits are highly polygenic. Thus, it is not expected that a single variant within a gene would be identified to drive these phenotypic results in humans. It is likely that small effects of multiple SNPs across multiple genes, including within the same gene, and with nonadditive effects (e.g., gene-by-environment effects, *Yen and Curran, 2016*), contribute to the resulting phenotypes. With this use of gene-wide association scanning approach, it is only possible to identify variants associated with overall effects. Without identifying a causal SNP, we can only aggregate available data to suggest what contributes to a biological pathway. For example, through exploitation of the publicly

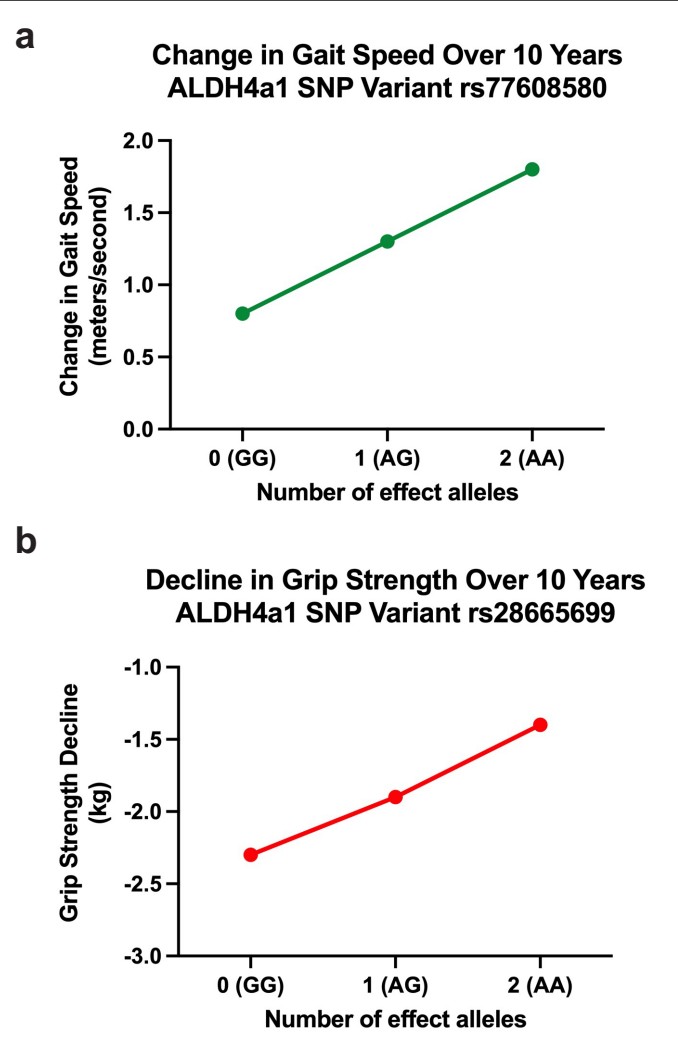

**Figure 3.** Effects of *ALDH4A1* variation on phenotypes representing association with change in aging-related function in a normative, population-based sample of older adults. (**a**) Change in gait speed over 10 years. Effect of SNP rs77608580 on aging-related changes in gait speed ($b = 0.052$, $p = 0.0025$). Over the span of one decade, on average, those with one or two effect alleles will have faster gait speeds with a difference of 0.52 and 1.04 m/s, respectively, compared to those without an effect allele. (**b**) Decline in grip strength over 10 years. Variation in *ALDH4A1* (SNP rs28665699) is inversely associated with decline in aging-related grip strength ($b = -0.045$, $p = 0.0009$). Individuals with one or two effect alleles have slower progression of weakened grip strength over 10 years by 0.5 and 1.0 kg, respectively, compared to the same aged individuals without the effect allele.

The online version of this article includes the following figure supplement(s) for figure 3:

**Figure supplement 1.** Association between rs77608580 and ALDH4A1 gene expression levels in whole blood.

available Genotype-Tissue Expression (GTEx) database (*Battle et al., 2017*), we found that one of the tag SNPs in *ALDH4A1*, rs77608580, was significantly associated with differential *ALDH4A1* expression levels through the association with an expression quantitative trait locus in whole blood (*Figure 3—figure supplement 1*). Further experimental studies to reveal the downstream effect(s) of altered gene expression and/or specific muscle functionality phenotyping, are required to address mechanistic questions pertaining to unique muscle groups and muscle-specific activities.

With the goal of better understanding the relationships between the phenotypes and potential disease, or system functionality, investigating more than one phenotype is an important strength (*Dey et al., 2017*; *Pendergrass et al., 2015*). Several studies have now demonstrated the biological utility of invoking multiple phenotypes for genetic association scans (*Hall et al., 2014*; *Pendergrass*

**Table 3.** Replication across ethnic subsamples in the HRS.

| Phenotype | SNP name | Location | Minor allele | European ancestry (N)* | African ancestry (N)* | Hispanic ancestry (N)* | Fixed effect p value† | Random effect p value‡ | Fixed effect†: OR or beta | Random effect‡: OR or beta | Q§ | I¶ |
|---|---|---|---|---|---|---|---|---|---|---|---|---|
| Grip strength decline | rs28665699 | 19200185 | A | 5228 | – | 409 | 0.00150 | 0.00150 | −0.0418 | −0.0418 | 0.3341 | 0.00 |
| Gait speed decline | rs77608580 | 19196968 | A | 3319 | 381 | 237 | 0.00775 | 0.72900 | 0.0424 | 0.0146 | 0.0577 | 64.95 |

*N: sample size by group included in the meta-analysis.

†Fixed effect: p value and effect size.

‡Random effect: p value and effect size.

§Cochrane's Q statistic: indicator of variance across sample effect sizes.

¶I: heterogeneity index to quantify dispersion across samples.

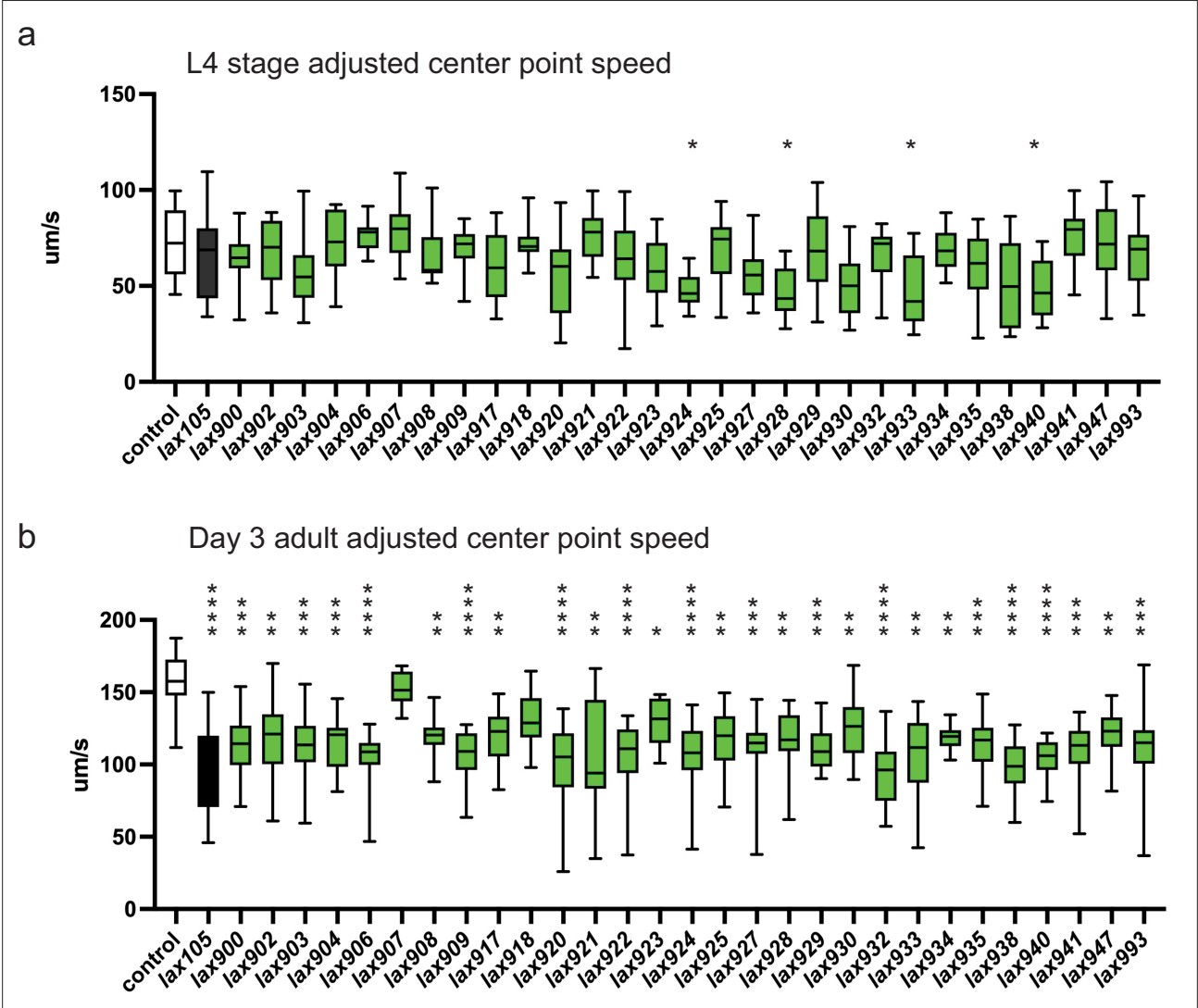

**Figure 4.** *alh-6* mutations accelerate loss of muscle function. WormLab software analysis of adjusted center point speed of individual animals of the given genotypes at the L4 stage (**a**) or day 3 of adulthood (**b**). Brown–Forsythe and Welch analysis of variance (ANOVA) test with Dunnett's T3 multiple comparisons test, with individual variances computed for each comparison. *p < 0.05; **p < 0.01; ***p < 0.001; ****p < 0.0001.

The online version of this article includes the following figure supplement(s) for figure 4:

**Figure supplement 1.** *alh-6* mutations accelerate loss of muscle function.

*et al., 2013*). Invoking more than one phenotype and multiple SNPs for genetic testing, however, brings forth new challenges when needing to consider multiple hypothesis-testing burden, or type 1 error, while not missing underlying associations from overly stringent significance criteria that typically assume independent genetic variants and phenotypes. Prior studies that have used a multiple pheno-type approach to investigate upwards of a thousand phenotypes do so with a completely agnostic design, used for exploratory hypothesis generation (*Hall et al., 2014*; *Pendergrass et al., 2013*). In contrast, with the current study, we sought to use the selected phenotypes in human data to cross-validate findings, with hypotheses generated from a model organism. Thus, the current study uses a more targeted, hypothesis-driven approach, by limiting a study to two selected phenotypes and variants within one identified genomic region. With this design, we retain the potential for detecting possible areas of genetic pleiotropy (i.e., genetic variation on more than one phenotype) and possib-lities for identifying mechanistic pathways leading to normative aging-related muscle functioning and decline.

Existing biomarkers of muscle health that can accurately predict muscle health later in life are extremely scarce due to limited data in human aging and an incomplete understanding of the molec-ular basis of sarcopenia. Moreover, facile approaches to experimentally validate the hypothesis gener-ated from deep human genetic variation datasets are scarce. Understanding the diversity of genetic variation underlying sarcopenia, as well as their corresponding phenotypic outcomes, will be critical for providing accurate risk assessments for family planning and genetic counseling of older adults. We have established a powerful experimental platform that synergistically utilizes the data rich resources of the US Health and Retirement study with the genetically tractable and methodologically rich *C. elegans* model. We anticipate this new research paradigm will be a formidable tool for collaboration between computational and bench scientists. Although the etiology of human disease is complex and multifactorial, we have used a combination of classical *C. elegans* genetics and human genetic association studies to define genetic variation in *alh-6/ALDH4A1* as a new biomarker of age-related muscle health in human.

## Materials and methods

### *C. elegans* strains and maintenance

*C. elegans* were cultured using standard techniques at 20°C (*Brenner, 1974*). The following strains were used: wild-type (WT) N2 Bristol, SPC321 [*alh-6(lax105)*], CL2166[*gst-4p::gfp*], SPC223 [*alh-6(lax105);gst-4p::gfp*], SPC542 [*alh-6(lax917);gst-4p::gfp*], SPC531 [*alh-6(lax906);gst-4p::gfp*], SPC528 [*alh-6(lax903);gst-4p::gfp*], SPC552 [*alh-6(lax927);gst-4p::gfp*], SPC561 [*alh-6(lax937);gst-4p::gfp*], SPC564 [*alh-6(lax940);gst-4p::gfp*], SPC549 [*alh-6(lax924);gst-4p::gfp*], SPC566 [*alh-6(lax945);gst-4p::gfp*], SPC540 [*alh-6(lax915);gst-4p::gfp*], SPC562 [*alh-6(lax938);gst-4p::gfp*], SPC563 [*alh-6(lax939);gst-4p::gfp*], SPC546 [*alh-6(lax921);gst-4p::gfp*], SPC527 [*alh-6(lax902);gst-4p::gfp*], SPC529 [*alh-6(lax904);gst-4p::gfp*], SPC536 [*alh-6(lax911);gst-4p::gfp*], SPC534 [*alh-6(lax909);gst-4p::gfp*], SPC559 [*alh-6(lax935);gst-4p::gfp*], SPC532 [*alh-6(lax907);gst-4p::gfp*], SPC569 [*alh-6(lax993);gst-4p::gfp*], SPC544 [*alh-6(lax919);gst-4p::gfp*], SPC562 [*alh-6(lax938);gst-4p::gfp*], SPC551 [*alh-6(lax926);gst-4p::gfp*], SPC530 [*alh-6(lax905);gst-4p::gfp*], SPC533 [*alh-6(lax908);gst-4p::gfp*], SPC548 [*alh-6(lax923);gst-4p::gfp*], SPC550 [*alh-6(lax925);gst-4p::gfp*], SPC565 [*alh-6(lax941);gst-4p::gfp*], SPC538 [*alh-6(lax913);gst-4p::gfp*], SPC543 [*alh-6(lax918);gst-4p::gfp*], SPC541 [*alh-6(lax916);gst-4p::gfp*], SPC545 [*alh-6(lax920);gst-4p::gfp*], SPC554 [*alh-6(lax929);gst-4p::gfp*], SPC558 [*alh-6(lax934);gst-4p::gfp*], SPC526 [*alh-6(lax901);gst-4p::gfp*], SPC568 [*alh-6(lax992);gst-4p::gfp*], SPC535 [*alh-6(lax910);gst-4p::gfp*], SPC553 [*alh-6(lax928);gst-4p::gfp*], SPC539 [*alh-6(lax914);gst-4p::gfp*], SPC556 [*alh-6(lax932);gst-4p::gfp*], SPC557 [*alh-6(lax933);gst-4p::gfp*], SPC567 [*alh-6(lax947);gst-4p::gfp*].

Double mutants were generated by standard genetic techniques. *E. coli* strains used were as follows: OP50/*E. coli* B for standard growth. All genetic mutants were backcrossed at least 4× prior to phenotypic analyses.

### Genetic complementation (cis–trans) testing

Hermaphrodites from each isolated mutant that phenocopied the *alh-6(lax105)*-like, age-related acti-vation of the *gst-4p::gfp* reporter in the musculature were mated to SPC223 [*alh-6(lax105);gst-4p::gfp*] males. F1 progeny were screened at day 3 of adulthood for the *alh-6(lax105)*-like phenotype, which

indicates a failure of the *alh-6(lax105)* allele to complement the mutation in the new mutant strain; thus the new mutant harbors a loss-of-function allele in *alh-6*.

### DNA sequencing of *alh-6* genetic mutants

Approximately 200 adult worms were collected and washed with M9. Animals were homogenized and genomic DNA was extracted using the Zymo Research Quick-DNA Miniprep kit (Cat. #D3025). The entire *alh-6* genomic sequence (ATG to stop) was amplified by PCR and cloned in a linearized pMiniT 2.0 vector (NEB PCR Cloning Kit, Cat. #E1202S). Plasmid DNA was purified using the Zymo Research Zyppy Plasmid Miniprep kit (Cat. D4019) and sequenced.

### Microscopy

Zeiss Axio Imager and ZEN software were used to acquire all images used in this study. For GFP reporter strains, worms were mounted in M9 with 10 mM levamisole and imaged with DIC and GFP filters. Worm areas were measured in ImageJ software (National Institutes of Health, Bethesda, MD) using the polygon tool.

### HRS human samples

The US HRS (*Sonnega et al., 2014*; *Juster and Suzman, 1995*) is a nationally representative, longitudinal sample of adults aged 50 years and older, who have been interviewed every 2 years, beginning in 1992. Because the HRS is nationally representative, including households across the country and the surveyed sample now includes over 36,000 participants, it is often used to calculate national prevalence rates for specific conditions for older adults, including physical and mental health outcomes, cognitive outcomes, as well as financial and social indicators.

The sample for the current study is comprised of a subset of the HRS for which genetic data were collected, as described below. To reduce potential issues with population stratification, the GeneWAS in this study was limited to individuals of primarily European ancestry. The final sample was $N = 3319$, with the proportion of women at 58.5%.

### Genotyping data

For HRS, genotype data were accessed from the National Center for Biotechnology Information Genotypes and Phenotypes Database (dbGaP; *HRS Health and Retirement Study, 2021*). DNA samples from HRS participants were collected in two waves. In 2006, the first wave was collected from buccal swabs using the Qiagen Autopure method (Qiagen, Valencia, CA). In 2008, the second wave was collected using Oragene saliva kits and extraction method. Both waves were genotyped by the NIH Center for Inherited Disease Research (CIDR; Johns Hopkins University) using the HumanOmni2.5 arrays from Illumina (San Diego, CA). Raw data from both phases were clustered and called together. HRS followed standard quality control recommendations to exclude samples and markers that obtained questionable data, including CIDR technical filters (*Laurie et al., 2010*), removing SNPs that were duplicates, had missing call rates ≥2%, >4 discordant calls, >1 Mendelian error, deviations from Hardy–Weinberg equilibrium (at p value $< 10^{-4}$ in European samples, and sex differences in allelic frequency ≥0.2). Further detail is provided in HRS documentation (*Weir, 2013*). Applying these criteria to the gene region, on chromosome 1, (NC_000001.10): 19,194,787–19,232,430 resulted in available data on 70 SNPs within the *ALDH4A1* region that are on the Illumina array to represent 273 human SNPs in the gene. With the goal of evaluating whether representative marker SNPs across the gene are associated with the phenotypes of interest, we implemented a pruning procedure, which sequentially scans SNPs in linkage disequilibrium, and performs thinning to subset to more independent SNPs based on a given threshold of correlation between SNPs and between linear combinations of SNPs. To achieve this, SNPs were first filtered to retain 53 SNPs that had a minor allele frequency at 0.01 or greater. We then pruned by recursively removing SNPs within a sliding window of 25 (i.e., 25 consecutive SNPs), shifted the window with 5 SNPs forward, and set the variance inflation factor threshold at 2. This yielded 21 SNPs for consideration (*Table 2*).

### Statistical analysis of HRS dataset

Following SNP extraction, we followed analytical steps of Phenotype construction GeneWASs, and SNP evauation.

## HRS phenotype construction

HRS phenotype construction was completed to calculate common measures of normal age-related muscle decline in functionality over time. *Figure 2—source data 1* shows the HRS data years from which phenotypes were calculated and details on how the variable is defined, and score or variable range. Datasets from multiple survey years were merged to get repeated assessments of variables on the same individuals. Phenotypes were calculated based on consensus following a review of the literature on assessments for age-related outcomes for variables implemented in the HRS and similar population-based surveys of aging. Further background for coding of specific phenotypes is described in detail previously for gait (*Wu et al., 2017*; *Batsis et al., 2016*; *Kim et al., 2019*). Phenotypes for grip strength decline and gait speed decline were assessed as change in performance on those tasks over time. Change was calculated by taking the score from the most recent assessment and subtracting the score from the first assessment for each person, within the respective years listed. Additional descriptive statistics on phenotypes can be provided. Phenotypes were calculated using SAS 9.4.

## GeneWAS

GeneWAS occurred through separate linear regression scans, under an additive model, adjusting for relevant covariates and indicators of population stratification as described below.

### Population stratification

As with any statistical analysis of association, if the correlation between dependent and independent variables differs for subpopulations, this may result in spurious genetic associations (*Novembre et al., 2008*). To reduce such type 1 error, we conducted the GeneWAS adjusting for population structure as indicated by latent factors from principal components analysis (PCA) (*Tian et al., 2008*; *Price et al., 2006*). Detailed descriptions of the processes employed for running PCA, including SNP selection, are provided by HRS, and follow methods outlined by *Price et al., 2006*. Two PCAs were run. The first PCA included 1230 HapMap anchors from various ancestries and was used to test against self-reported race and ethnic classifications. Several corrections to the dataset were made based on this analysis. The second PCA was run on the corrected dataset, on unrelated individuals and excluding HapMap anchors, to create eigenvectors to serve as covariates and adjust for population stratification in association tests. From the second PCA, the first two eigenvalues with the highest values accounted for less than 4.5% of the overall genetic variance, with additional components (3–8) increasing this minimally, by a total of ~1.0% (*Weir, 2013*). Based on these analyses, we opted for a strategy that does not ignore population substructure, but also does not overcorrect, and adjusted for the first four PCs in all analyses. When coupling this approach of adjusting for principal components with all quality control procedures performed, excluding any related individuals and limiting the dataset for ancestral homogeneity, we reduce the likelihood of false associations resulting from population stratification (*Tian et al., 2008*; *Price et al., 2006*; *Li and Yu, 2008*; *Serre et al., 2008*; *Zhang et al., 2003*; *Zhu et al., 2002*; *Price et al., 2010*).

### Regression models and other covariates

When conducting regressions on phenotypes indicating change over time, additional adjustments were made using covariates for baseline levels, number of years during which change was calculated, and variables shown to affect outcomes. For example, with change in gait speed, a linear regression scan was run adjusting for sex, age at the first assessment point, number of years of follow-up, baseline walking speed, and floor type in addition to principal components. All GeneWASs were completed using PLINK 2.0 (*Purcell et al., 2007*). The strength of the associations, as indicated by effect sizes and p values, is not directly comparable for each phenotype because the sample sizes differed by phenotype. Thus, the strength of an association does not reflect how strong a SNP effects one phenotype compared to another. Because we did find more than one variant associated with the phenotypes, we are more confident that these results were not due to type 1 error.

## SNP evaluation

We evaluated SNP associations in the GeneWAS by p value. With the number of SNPs and primary phenotypes in this study, strict Bonferroni correction would yield an adjusted multiple test correction

p value threshold of 0.0012 (for 21 × 2 tests). However, Bonferroni correction such as these are too conservative because of the correlations among SNPs (*Han et al., 2009*; *Sham and Purcell, 2014*) and the cross-validation approach. To address the correlation among SNPs, we implement a pruning schema and calculate empirical p value thresholds, through permutation (*Han et al., 2009*; *Sham and Purcell, 2014*; *Dudbridge and Gusnanto, 2008*; *Pahl and Schäfer, 2010*). Permutation is a process whereby necessary correlations between SNPs and phenotypes are intentionally shuffled so that p values for the shuffled (null) data are compared to the nonshuffled data. This permutation is repeated multiple times in order to determine an empirical p value (*Sham and Purcell, 2014*; *Pahl and Schäfer, 2010*; *North et al., 2002*), a calculated threshold at which a test result is less likely to achieve significance by chance alone. Thus, when performing 1000 permutations using PLINK and max(T) option (*Pahl and Schäfer, 2010*), the empirical p value thresholds of 0.0019 for grip strength decline and 0.006 for gait speed decline were observed for determining gene-wide significance. For SNP comparisons, we used R (CRAN; https://www.r-project.org).

## Gene expression

The GTEx database (*Battle et al., 2017*), the most comprehensive, publicly available resource for tissue-specific gene expression data, was used to evaluate whether there was evidence for regulatory functions of SNPs within the gene. We entered the top SNPs into GTEx to assess relationships with differential gene expression.

## WormLab measurements

As previously described (*Roussel et al., 2014*), but in brief; 15–20 animals were moved to a NGM stock plate without *E. coli* OP50 and recorded in WormLab software (MBF Bioscience) for 2 min.

## Swimming measurements

As previously described (*Stuhr and Curran, 2020*), but in brief; 15–20 worms were moved to an unseeded NGM stock plate for 1 hr. Then worms were washed with M9 into 5 µl drops onto a fresh NGM plate. After 1 min, 15–20 worms were imaged via Movie Recorder at 50 ms exposure using ZEN 2 software (Zeiss Axio Imager).

## Statistical analysis of *Alh-6* genetic mutants

Data are presented as mean ± SD. Comparisons and significance were analyzed in GraphPad Prism 8. Comparisons between more than two groups were done using analysis of variance.

## Acknowledgements

We thank J Gonzalez for technical assistance, Dr. W Mack for statistical consultation, and Drs. R Irwin and C Duangjan for critical reading of the manuscript. Some strains were provided by the CGC, which is funded by the NIH Office of Research Infrastructure Programs (P40 OD010440). This work was funded by the NIH R01 AG058610 and RF1 AG063947 to SPC, T32 AG052374 to OV, and NLS and T32 GM118289 to NLS. This study was supported in part by funding from The National Institute on Aging, through the USC-Buck Nathan Shock Center (P30 AG068345). The National Institute on Aging has supported the collection of both survey and genotype data for the Health and Retirement Study through co-operative agreement U01 AG009740. The datasets are produced by the University of Michigan, Ann Arbor. The HRS phenotypic data files are public use datasets, available through: https://hrs.isr.umich.edu/data-products/access-to-public-data. The HRS genotype data are available to authorized researchers: https://www.ncbi.nlm.nih.gov/projects/gap/cgi-bin/study.cgi?study_id=phs000428.v2.p2https://www.ncbi.nlm.nih.gov/projects/gap/cgi-bin/study.cgi?study_id=phs000428.v2.p2.

## Additional information

### Funding

| Funder | Grant reference number | Author |
|---|---|---|
| National Institute on Aging | R01 AG058610 | Sean P Curran |
| National Institute on Aging | RF1 AG063947 | Sean P Curran |
| National Institute on Aging | T32 AG052374 | Osvaldo Villa<br>Nicole L Stuhr |
| National Institute of General Medical Sciences | T32 GM118289 | Nicole L Stuhr |
| National Institute on Aging | P30 AG068345 | Sean P Curran<br>Thalida Em Arpawong |

The funders had no role in study design, data collection, and interpretation, or the decision to submit the work for publication.

### Author contributions

Osvaldo Villa, Formal analysis, Investigation, Methodology, Visualization, Writing - original draft; Nicole L Stuhr, Formal analysis, Investigation, Methodology, Visualization, Writing - review and editing; Chia-an Yen, Investigation, Methodology, Visualization, Writing - review and editing; Eileen M Crimmins, Data curation, Methodology, Resources; Thalida Em Arpawong, Data curation, Formal analysis, Methodology, Resources, Visualization, Writing - review and editing; Sean P Curran, Conceptualization, Formal analysis, Funding acquisition, Investigation, Methodology, Project administration, Supervision, Writing - original draft, Writing - review and editing

### Author ORCIDs

Osvaldo Villa http://orcid.org/0000-0002-0803-0752
Nicole L Stuhr http://orcid.org/0000-0003-2537-7114
Thalida Em Arpawong http://orcid.org/0000-0001-9671-9535
Sean P Curran http://orcid.org/0000-0001-7791-6453

### Decision letter and Author response

Decision letter https://doi.org/10.7554/eLife.74308.sa1
Author response https://doi.org/10.7554/eLife.74308.sa2

## Additional files

### Supplementary files

• Transparent reporting form

### Data availability

All data are available within the manuscript. Health and retirement study (HRS) data are maintained at the University of Michigan - https://hrs.isr.umich.edu/about.

The following dataset was generated:

| Author(s) | Year | Dataset title | Dataset URL | Database and Identifier |
|---|---|---|---|---|
| David W | 2017 | Health and Retirement Study (HRS) | https://www.ncbi.nlm.nih.gov/projects/gap/cgi-bin/study.cgi?study_id=phs000428.v2.p2 | NCBI Gene Expression Omnibus, phs000428.v2.p2 |

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

# Appendix 1

## Appendix 1—key resources table

| Reagent type (species) or resource | Designation | Source or reference | Identifiers | Additional information |
|---|---|---|---|---|
| Strain, strain background (*C. elegans*) | N2 | *Caenorhabditis* Genetics Center (CGG) | | Laboratory reference strain (wild type) |
| Strain, strain background (*C. elegans*) | SPC321 | PMID:24440036 | | Genotype: *alh-6(lax105)* |
| Strain, strain background (*C. elegans*) | CL2166 | *Caenorhabditis* Genetics Center (CGG) | | Genotype: *gst4-p::gfp* |
| Strain, strain background (*C. elegans*) | SPC223 | PMID:24440036 | | Genotype: *alh-6(lax105);gst-4p::gfp* |
| Strain, strain background (*C. elegans*) | SPC542 | This paper | | Genotype: *alh-6(lax917);gst-4p::gfp* |
| Strain, strain background (*C. elegans*) | SPC531 | This paper | | Genotype: *alh-6(lax906);gst-4p::gfp* |
| Strain, strain background (*C. elegans*) | SPC528 | This paper | | Genotype: *alh-6(lax903);gst-4p::gfp* |
| Strain, strain background (*C. elegans*) | SPC552 | This paper | | Genotype: *alh-6(lax927);gst-4p::gfp* |
| Strain, strain background (*C. elegans*) | SPC561 | This paper | | Genotype: *alh-6(lax937);gst-4p::gfp* |
| Strain, strain background (*C. elegans*) | SPC564 | This paper | | Genotype: *alh-6(lax940);gst-4p::gfp* |
| Strain, strain background (*C. elegans*) | SPC549 | This paper | | Genotype: *alh-6(lax924);gst-4p::gfp* |
| Strain, strain background (*C. elegans*) | SPC566 | This paper | | Genotype: *alh-6(lax945);gst-4p::gfp* |
| Strain, strain background (*C. elegans*) | SPC540 | This paper | | Genotype: *alh-6(lax915);gst-4p::gfp* |
| Strain, strain background (*C. elegans*) | SPC562 | This paper | | Genotype: *alh-6(lax938);gst-4p::gfp* |
| Strain, strain background (*C. elegans*) | SPC563 | This paper | | Genotype: *alh-6(lax939);gst-4p::gfp* |
| Strain, strain background (*C. elegans*) | SPC546 | This paper | | Genotype: *alh-6(lax921);gst-4p::gfp* |
| Strain, strain background (*C. elegans*) | SPC527 | This paper | | Genotype: *alh-6(lax902);gst-4p::gfp* |
| Strain, strain background (*C. elegans*) | SPC529 | This paper | | Genotype: *alh-6(lax904);gst-4p::gfp* |

*Appendix 1 Continued on next page*

*Appendix 1 Continued*

| Reagent type (species) or resource | Designation | Source or reference | Identifiers | Additional information |
|---|---|---|---|---|
| Strain, strain background (*C. elegans*) | SPC536 | This paper | | Genotype: *alh-6(lax911);gst-4p::gfp* |
| Strain, strain background (*C. elegans*) | SPC534 | This paper | | Genotype: *alh-6(lax909);gst-4p::gfp* |
| Strain, strain background (*C. elegans*) | SPC559 | This paper | | Genotype: *alh-6(lax935);gst-4p::gfp* |
| Strain, strain background (*C. elegans*) | SPC532 | This paper | | Genotype: *alh-6(lax907);gst-4p::gfp* |
| Strain, strain background (*C. elegans*) | SPC569 | This paper | | Genotype: *alh-6(lax993);gst-4p::gfp* |
| Strain, strain background (*C. elegans*) | SPC544 | This paper | | Genotype: *alh-6(lax919);gst-4p::gfp* |
| Strain, strain background (*C. elegans*) | SPC562 | This paper | | Genotype: *alh-6(lax938);gst-4p::gfp* |
| Strain, strain background (*C. elegans*) | SPC551 | This paper | | Genotype: *alh-6(lax926);gst-4p::gfp* |
| Strain, strain background (*C. elegans*) | SPC530 | This paper | | Genotype: *alh-6(lax905);gst-4p::gfp* |
| Strain, strain background (*C. elegans*) | SPC533 | This paper | | Genotype: *alh-6(lax908);gst-4p::gfp* |
| Strain, strain background (*C. elegans*) | SPC548 | This paper | | Genotype: *alh-6(lax923);gst-4p::gfp* |
| Strain, strain background (*C. elegans*) | SPC550 | This paper | | Genotype: *alh-6(lax925);gst-4p::gfp* |
| Strain, strain background (*C. elegans*) | SPC565 | This paper | | Genotype: *alh-6(lax941);gst-4p::gfp* |
| Strain, strain background (*C. elegans*) | SPC538 | This paper | | Genotype: *alh-6(lax913);gst-4p::gfp* |
| Strain, strain background (*C. elegans*) | SPC543 | This paper | | Genotype: *alh-6(lax918);gst-4p::gfp* |
| Strain, strain background (*C. elegans*) | SPC541 | This paper | | Genotype: *alh-6(lax916);gst-4p::gfp* |
| Strain, strain background (*C. elegans*) | SPC545 | This paper | | Genotype: *alh-6(lax920);gst-4p::gfp* |
| Strain, strain background (*C. elegans*) | SPC554 | This paper | | Genotype: *alh-6(lax929);gst-4p::gfp* |

*Appendix 1 Continued on next page*

*Appendix 1 Continued*

| Reagent type (species) or resource | Designation | Source or reference | Identifiers | Additional information |
|---|---|---|---|---|
| Strain, strain background (*C. elegans*) | SPC558 | This paper | | Genotype: *alh-6(lax934);gst-4p::gfp* |
| Strain, strain background (*C. elegans*) | SPC526 | This paper | | Genotype: *alh-6(lax901);gst-4p::gfp* |
| Strain, strain background (*C. elegans*) | SPC568 | This paper | | Genotype: *alh-6(lax992);gst-4p::gfp* |
| Strain, strain background (*C. elegans*) | SPC535 | This paper | | Genotype: *alh-6(lax910);gst-4p::gfp* |
| Strain, strain background (*C. elegans*) | SPC553 | This paper | | Genotype: *alh-6(lax928);gst-4p::gfp* |
| Strain, strain background (*C. elegans*) | SPC539 | This paper | | Genotype: *alh-6(lax914);gst-4p::gfp* |
| Strain, strain background (*C. elegans*) | SPC556 | This paper | | Genotype: *alh-6(lax932);gst-4p::gfp* |
| Strain, strain background (*C. elegans*) | SPC557 | This paper | | Genotype: *alh-6(lax933);gst-4p::gfp* |
| Strain, strain background (*C. elegans*) | SPC567 | This paper | | Genotype: *alh-6(lax947);gst-4p::gfp* |
| Sequence-based reagent | pMiniT 2.0 vector and cloning kit | New England Biolabs | #E1202S | |
| Software, algorithm | MBF Bioscience Wormlab | https://www.mbfbioscience.com/wormlab | | |
| Other | US Health and Retirement Study (HRS) | https://www.ncbi.nlm.nih.gov/projects/gap/cgi-bin/study.cgi?study_id=phs000428.v2.p2 | | National Center for Biotechnology Information Genotypes and Phenotypes Database **dbGaP Study Accession:** phs000428.v2.p2 |
| Strain, strain background (*Escherichia coli*) | OP50-1 | *Caenorhabditis* Genetics Center (CGG) | RRID:WB-STRAIN:WBStrain00041971 | Standard *E. coli* B diet Streptomycin resistant |
| Software, algorithm | GraphPad Prism | GraphPad Prism (https://graphpad.com) | RRID:SCR_015807 | Version 6 |
| Software, algorithm | ImageJ | ImageJ (http://imagej.nih.gov/ij/) | RRID:SCR_003070 | |
| Software, algorithm | Phyre2 | http://www.sbg.bio.ic.ac.uk/phyre2/html/page.cgi?id=index | | |
| Software, algorithm | MisSense3D | http://missense3d.bc.ic.ac.uk/missense3d/ | | |

