## [Editor Report]

Age-associated muscle decline is a pervasive aspect of human aging biology. Here the authors provide substantial genetic evidence implicating *C. elegans* mitochondrial enzyme ALH-6 (P5C dehydrogenase) in muscle oxidative stress and in the maintenance of muscle functionality late into life. The authors also analyzed databases on human aging to identify a linkage of human ALH-6 homolog ALDH4A1 and indicators of human muscle function in aging, which suggests conserved function of ALH-6/ALDH4A1 in aging and the potential for use of ALDH4A1 genetic data as a predictor for old age muscle health.

---

## [Decision Letter]

**Decision letter after peer review:**

Thank you for submitting your article "Genetic variation in *ALDH4A1* predicts muscle health over the lifespan and across species" for consideration by *eLife*. Your article has been reviewed by 3 peer reviewers, and the evaluation has been overseen by a Reviewing Editor and Carlos Isales as the Senior Editor. The following individual involved in review of your submission has agreed to reveal their identity: Alfred Fisher (Reviewer #3).

The reviewers raised important points regarding statistical significance, and the relationship of SNPs to ALDH41A expression/stability.

Following the suggestions in the reviews should enhance the readability and accuracy of presentation.

High priority-Essential

1) Please make a more compelling case for the significance of the identified SNPs. Reviewer 2 (R2) suggests a threshold for significance for the SNPs of p=5.8e-5, which is not cleared by reported SNPs.

More specifically, R2 says: "The authors tested the significance of 53 SNPs in the ALDH4A1 locus with 16 phenotypes in the HRS (Table S1). They claim that they are performing a gene-wide association study and are correcting for 16 tests (0.05/16) in their statistical significance assessment. This analysis is not correct. The authors need to correct for 53x16=848 tests (0.05/848 = 5.8e-5 if they choose to use Bonferroni correction). The authors can argue that the SNPs are not independent, which would be true. In that case, they need to prune the SNPs based on linkage disequilibrium and use index/tag SNPs for their analyses. Further, some of the 16 phenotypes could also be highly correlated; and this needs to be acknowledged/addressed".

– the authors should justify the P-value threshold with strong statistical argument or a reinforcing analysis. If this is not possible, the work as presented here cannot be recommended as adequately rigorous.

If possible, address replication in cohorts. R3 suggests that a validation of the importance could involve author identification of the best phenotype and SNPs (as they have done), and then to validate these findings in other muscle aging data sets. The authors could use a similar approach to this work, but with many fewer SNPs and phenotypes, which would lower the p-value threshold, and make the work more robust. Such an approach would exploit existing electronic data, so the work involved should be fairly modest.

2) Please link SNPs to ALDH4 expression or stability to provide some sense of how a SNP could result in functional changes--making this connection compelling underlies the major conclusion of the paper.

We suggest that you mine the eQTL and RNA stability databases to determine which SNPs can be linked to some potential change in ALDH41A expression/stability. The eQTL analysis is both important to substantiate the claims and doable. What the authors would need to do to substantiate their claim that ALDH41A is a biomarker of muscle health in humans is to mine and perform the appropriate genetics and statistical analyses using existing cell-specific data in the extensive human eQTLs repositories (e.g., GTEx, eQTLGen, etc). Tissue-specific human eQTL data are publicly available, and can be extracted and analyzed. For more information on these repositories, the authors can refer to PMID: 34493866.

Better connect predictor human SNPs and *C. elegans* muscle aging biology.

The SNPs of human ALDH4A1 were not analyzed with respect to the *C. elegans* alh-6 mutations. Reviewers agreed that making a "functional" connection from nematode to human SNP would strengthen the arguments put forward in the paper. Minimally, the authors should include clear listing of whether SNPs might be associated with a change in the *C. elegans* protein/transcript.

Easy additions for clarity and value:

The goal of establishing a functional link between identified SNPs and ALDH4A1 function in human muscle can and should be enhanced by adding details regarding SNP impact on protein structure and function prediction, or of potential mRNA consequence (splicing site perturbation). Compare conserved amino acid sequences of *C. elegans* ALH-6 and human ALDH4A1 in parallel. Include schematic illustration of ALH-6 protein and predicted structures of ALH-6 wild-type and mutant proteins using protein structure prediction tools (i.e. AlphaFold2), which will provide useful data for alh-6 substitutions identified.

Suggested but not essential:

Constructing human cognate CRISPR alleles in nematodes would be welcome (especially if any support a clear structure/function hypothesis), but since 1) it is possible that a SNP change might not directly correspond to nematode impact even if the proposed relationship were operative, and 2) such engineering could require a fair amount of elegans manipulation, making this connection is not considered a requirement for a successful revision.

3) Comment on novelty of approach. The GWAS mining approach is important as it reports a success that opens up a novel avenue for connecting *C. elegans* biology/genetics to human physiology. Still, success with this approach for human obesity genes have been recently reported (PLoS Genetics; PMID: 34492009), so the approach is not the first of its type. Authors should cite that literature; their contribution on this front is still significant and the application in this system moves this approach toward a more central activity in the field, which is powerful.

4) The title should be precise and true to findings. R1 noted that "The title of this paper is somewhat misleading. The authors produced a prediction model of age-dependent decline of muscle functions based on genotypes of human ALDH4A1, but not those of *C. elegans* alh-6". The title. e.g., "Genetic variation in ALDH4A1 is associated with muscle health over the lifespan and across species" should be revised for accuracy.

5) Explain better, write carefully on SKN-1 activity claims. R2 notes that gst-4::GFP can be induced by SKN-1 as well as by additional transcription factors; SKN-1 activity is not directly tested in this study. Authors should directly test SKN-1 activation, or just state gst-4::GFP expression as the outcome assayed, with caveats mentioned. The authors have previously produced data on skn-1 in this response, possibly more extensive discussion of that data might allay some of the concerns.

Authors should also consider R1 points about muscle-specific activation of gst-4::gfp claims.

In sum, in revision, the authors precise as to what is directly assayed in the screen (gst-4::GFP expression, not necessarily SKN-1 activity) in the summary.

*Reviewer #1 (Recommendations for the authors):*

1. I think the biggest issue for this excellent paper is that the information about SNPs of human ALDH4A1 was not analyzed with respect to the *C. elegans* alh-6 mutations. Do the SNPs occur at the same or similar loci of the orthologous *C. elegans* mutation sites? If not, can the authors introduce human SNPs into *C. elegans* for the orthologous changes and test whether they affect age-dependent declines in muscle functions? This is the key for improving the paper to fit the purpose of the work.

2. The title of this paper is somewhat misleading. The authors produced a prediction model of age-dependent decline of muscle functions based on genotypes of human ALDH4A1, but not those of *C. elegans* alh-6. Please downplay the title. e.g., "Genetic variation in ALDH4A1 is associated with muscle health over the lifespan and across species"

3. In Figure 1b, please show schematic illustration of ALH-6 protein. In addition, predicted structures of ALH-6 wild-type and mutant proteins using protein structure prediction tools (i.e. AlphaFold2) will provide useful information for alh-6 mutations (for example, "G152/K153" in page 5 line 89).

4. In Figure 1c, please describe accurately how the authors executed One-way ANOVA statistical analysis for measuring GFP intensity.

5. I recommend comparing conserved amino acid sequences of *C. elegans* ALH-6 and human ALDH4A1 in parallel.

6. In figure supplement 1, please add a scale bar.

7. In Figure 2, please elaborate the choice of p values 0.003 and 0.013.

8. In Figure 3, please mark exact genotypes at x axes.

9. For Figure 3, p values are in written the Figure Legends and Table 1. However, it will be better to mark p values or asterisks in the panels as well.

10. I recommend adding discussions regarding how genetic variants in alh-6/ALDH4A1 contribute to age-dependent impairment in muscle functions. In particular, it will be great to add speculation for changes in mitochondrial proline dehydrogenase levels and/or activities that may affect sarcopenia.

11. Please change one of "IADLA Decline" to "IADLA2 Decline" in Figure 2.

12. On page 5, line 85, add lax918 in addition to lax903 and lax930 for E78Stop.

13. On page 6, line 163, use "day 3" instead of "Day 3" for consistency.

*Reviewer #2 (Recommendations for the authors):*

To strengthen the main claims this reviewer suggests a revision of the following points:

1) The authors state:

"Based on the striking specificity of the muscle-restricted and age-dependent activation of SKN-1 in *C. elegans* harboring mutations in alh-6,…"

However, currently, data showing SKN-1 activation in the alh-6 mutant backgrounds are lacking. Direct observation of SKN-1::GFP activation would be important because the transcriptional reporter gst-4P::GFP, although a target of SKN-1, is also activated by other stress-responsive TFs including NHR-49 (PMID: 30297383) and DAF-16 (PMID: 32161088).

In addition, in Suppl. Figure 1 various alh-6 mutant alleles show non-muscle gst-4P::GFP signal.

Therefore, the authors may need to directly test SKN-1 activation, and revise the muscle-specific claims.

2) Structural predictions may enable the authors to generate hypotheses suggesting how a very early STOP codon in alh-6 (e.g. lax906) may have a phenotypic impact on muscle function comparable to a late amino acid substitution (e.g. lax947).

3) The authors tested the significance of 53 SNPs in the ALDH4A1 locus with 16 phenotypes in the HRS (Table S1). They claim that they are performing a gene-wide association study and are correcting for 16 tests (0.05/16) in their statistical significance assessment. This analysis is not correct. The authors need to correct for 53x16=848 tests (0.05/848 = 5.8e-5 if they choose to use Bonferroni correction). The authors can argue that the SNPs are not independent, which would be true. In that case, they need to prune the SNPs based on linkage disequilibrium and use index/tag SNPs for their analyses. Further, some of the 16 phenotypes could also be highly correlated; and this needs to be acknowledged/addressed.

4) It is unclear how the SNPs presented in Table 1 are linked to ALDH4A1. The SNPs are in the locus, but that does not necessarily mean they have a functional impact on ALDH4A1. For instance, do the identified SNPs lead to amino acid, splicing, or other regulatory changes in ALDH4A1?

Ultimately, this paper aims to advance ALDH4A1 as genetic factor underlying muscle-function decline in aging humans. Therefore, to go beyond the already publicly available link between the SNPs in the ALDH4A1 locus and muscle performance in aging humans, the authors may need to define whether the reported SNPs are linked to ALDH4A1 expression as eQTLs, especially in muscle tissue/cells.

5) For clarity, the distribution of the phenotypes for each allele may be presented in Figure 3 like they are presented in Figure 4.

*Reviewer #3 (Recommendations for the authors):*

1) The finding that alh-6 mutants exhibit declines in mobility during aging could reflect a selective effect on muscle function, or could be reflective of a larger acceleration of aging. It would be helpful to show lifespan data for the alh-6 mutant, or discuss if this work has been published previously.

2) The conclusion that the alh-6 mutant affects muscle function during aging could be further bolstered via studying muscle mass and structure as well as mitochondrial number and structure will readily available reporters.

3) For the human SNP analysis, engaging the help of an expert in human Kinesiology or Geriatrics can help with narrowing the phenotypes to make them more selective for muscle function, and exclude those where the connection to muscle function is more tangential.

4) Several recent papers have used genome-wide association analysis to identify SNPs associated with decreased muscle strength in middle-aged and older individuals. This work will provide information on other readily available dataset that could be suitable for a replication study to determine if the findings are seen in separate study populations.

5) For the SNP studies, the 0.003 p-value cut-off is likely too low given that 53 SNPs are being studied making the multiple testing greater than the correction used.

[Editors’ note: further revisions were suggested prior to acceptance, as described below.]

Thank you for resubmitting your work entitled "Genetic variation in ALDH4A1 is associated with muscle health over the lifespan and across species" for further consideration by *eLife*. Your revised article has been evaluated by Carlos Isales (Senior Editor), a Guest Reviewing Editor, the original reviewers, and an expert in statistics. We are sorry about the delay.

While Reviewer 1 is now in favor of acceptance as is, Reviewer 3 states: "I still have concerns about the multiple testing correction being used by the authors, and I did not feel that the authors addressed this in a fully compelling manner, such as adding a statistician."

Reviewer 2 concurs: "Although I would really like for the work to be complete, I'd need to agree with Reviewer 3. In addition to unjustified processing of the data that leads to a significance level (0.05) that seems inappropriate for this type of study, the authors have not addressed the concerns related to experimental testing of some of the key claims in *C. elegans*. For instance, the authors did not directly look at SKN-1::GFP even though the tools are available and this is a feasible experiment. As explained before, the current readout is indirect (gst-4::GFP), and although gst-4 is a target of SKN-1, it is also a target of DAF-16 and other transcription factors. Therefore, gst-4::GFP levels are only suggestive of SKN-1 activation. I agree that adding a GWAS expert statistician would be really helpful."

*eLife* consultant on statistics questions added: "It is true that in candidate gene association you may not need to correct for multiple tests, however, this is only the case when SNPs used are known to have a function (usually non-synonymous coding). I support the suggestion made by reviewers to use linkage disequilibrium (LD) for SNP pruning and also reduce the number of phenotypes tested if possible.Regarding SNPs, the authors' suggestion that all 53 SNPs are "tags" of unknown functional variants is not justified. How were they selected? If they are correlated, many of them still do tag the same function. Whether pruned or not, SNP significant association from this analysis is only an association, not causality especially given the complex nature of the phenotypes. It is true that LD pruning would make it difficult to replicate findings in other studies, particularly from other populations but it would be possible to use the nearest SNP available in those studies (as would also be the case if a different genotyping array was used).

Regarding phenotypes, first of all it is good to have such a richly annotated data. However, as already acknowledged, these are not independent and correcting for 16 tests is unduly self-penalising (just as it would be to correct for 53 SNPs). It would be more appropriate to set one phenotype (or a few unrelated) as the primary focus and others as exploratory or explanatory of the main ones if a link can be established between them (which I think there is)."

Please address these issues in what will hopefully be the last round of revision.

---

## [Author Response]

High priority-Essential1) Please make a more compelling case for the significance of the identified SNPs. Reviewer 2 (R2) suggests a threshold for significance for the SNPs of p=5.8e-5, which is not cleared by reported SNPs.More specifically, R2 says: "The authors tested the significance of 53 SNPs in the ALDH4A1 locus with 16 phenotypes in the HRS (Table S1). They claim that they are performing a gene-wide association study and are correcting for 16 tests (0.05/16) in their statistical significance assessment. This analysis is not correct. The authors need to correct for 53x16=848 tests (0.05/848 = 5.8e-5 if they choose to use Bonferroni correction). The authors can argue that the SNPs are not independent, which would be true. In that case, they need to prune the SNPs based on linkage disequilibrium and use index/tag SNPs for their analyses. Further, some of the 16 phenotypes could also be highly correlated; and this needs to be acknowledged/addressed".– the authors should justify the P-value threshold with strong statistical argument or a reinforcing analysis. If this is not possible, the work as presented here cannot be recommended as adequately rigorous.If possible, address replication in cohorts. R3 suggests that a validation of the importance could involve author identification of the best phenotype and SNPs (as they have done), and then to validate these findings in other muscle aging data sets. The authors could use a similar approach to this work, but with many fewer SNPs and phenotypes, which would lower the p-value threshold, and make the work more robust. Such an approach would exploit existing electronic data, so the work involved should be fairly modest.

These are critical issues, and we thank the reviewers for bringing them up so we can add clarity to the approach. We did not prune SNPs based on linkage disequilibrium (LD) for several reasons. First, to address SNP pruning, it is unknown which SNPs represent conservation across species and given unique human ancestral lineages, we did not want to miss potential SNP associations that represent important variation within the gene. It is now known that not all SNPs function equally across a gene, rather such things as positioning (exons closer to 5’ UTR, exons closer to 3’ UTR, introns) show different enrichment patterns within a gene, and thus effects can reflect differently in association studies; these differences are not clearly differentiated by LD between SNPs. Secondly, because each of the 53 SNPs within the gene represents a tag, or marker SNP for human variation within the gene with this association testing, we are not able to identify a causal variant, rather we are only to index that some variation within the gene suggests conserved associations with humans. Given this study design, we did not expect to identify causal SNPs, but rather sought to establish a pattern of association across all SNPs. So, by not pruning, if SNPs within our array are suggestive of association with phenotypes, yet are tagging SNPs not represented in the array, we would not miss further evaluating the SNP associations to represent variation within the gene particularly if suggestive associations appeared across more than one phenotype (which we now make clearer the correlations that exist between them).

Relatedly, with regard to the multiple phenotypes, the traits used in these scans came from a population-based study so that the traits were not assessed to allow us to identify physiological degeneration in specific muscles, rather to index and track overall age-related decline in functionality over time. It is widely accepted now that the genetic variation underlying these aging-related traits are highly polygenic. Thus, it is not expected that a single variant within a gene would be identified to drive these phenotypic results in humans. It is likely that small effects of multiple SNPs across multiple genes, including within the same gene, and with non-additive effects (e.g., gene-by-environment effects), contribute to the resulting phenotype. With this use of gene-wide association scanning approach, it is only possible to identify those variants contributing to overall effects. This is, for example, one reason SNPs are not always pruned for LD in the creation of polygenic risk scores, another being that pruning would be highly skewed towards the dominant ancestry sample. Also related, LD structures are explicitly tied to ancestry among humans (e.g., distinct demographic and recombination histories of groups). Thus, as mentioned, pruning by LD would be specific to the human cohort for this study, of primarily European ancestry and for future replication in cohorts, any pruning would potentially remove SNPs that are more important in cohorts with different LD structures. Because we are validating a finding initially resulting from *C. elegans*, there would be no rationale for pruning SNPs based on the LD structure of a single ancestry cohort. In the text, we now make better acknowledgements to the application of these techniques and the goals for these genewide-scans that invoked all SNPs. Given this approach, we did not calculate a p-value threshold on the basis of the number of SNPs rather designed the scan using all SNPs as markers for capturing variation within the single gene (Bonferroni corrected p-value of 0.0031 = 0.05/16 total phenotypes for variation in one gene). We now emphasize in the text that with these results among humans, we cannot identify a causal SNP nor elucidate a mechanism for the biological pathway, but can validate that there is an association between variation within the gene with muscle functionality that was previously found in *C. elegans*. Further experimental studies, or specific muscle functionality phenotyping, would be required to address the mechanisms.

Also, thank you for pointing out this important aspect of the approach with regard to correlated phenotypes. The 16 phenotypes selected for testing are correlated by design, with the aim of establishing a pattern across aging-related muscle phenotypes that represent aging-related physical functioning, in a normative aging sample. (We did not intend to find a single genetic variant that would explain functionality of a specific muscle group responsible for walking, for example.) Specifically, we chose to include common composite measures of physical functioning that index aging-related decline, including ADLA, IADLs (3 and 5 item composites), mobility, and large muscle functioning. Because we viewed these composite phenotypes as sensitivity measures, and ones that subsume other individual phenotypes tested, we did not consider these to be independent. For example, the ADLA composite includes 5 tasks (bathing, eating, dressing, walking across a room, and getting in or out of bed) and as shown, the SNP association with the individual phenotype of walking across a room was stronger than for the composite ADLA. With IADL decline phenotypes, we show these as broader composites that indicate how changes in human functionality are associating with gene variation in addition to the individual phenotypes that show weaker and suggestive associations (e.g., getting up from a chair, jogging, lifting 10 lbs, walking 1 block, walking upstairs). Thus, we emphasize that the inclusion of the composites as overlapping phenotypes aid in establishing a pattern. Relatedly, we did not include composite phenotypes in the calculation of the suggestive p-value cut-off and grouped other phenotypes by physiological functionality with arm pushing/pulling, getting up from a chair, walking and jogging, and grip phenotypes (0.5/4=0.013) with the approach of testing for associations among SNPs representing variation across the gene.

With regard to replication in other cohorts, access to harmonized phenotypes and genetic data is required; however, we were able to more immediately test replication across ethnic subsamples in the HRS by calculating a common effect size across the samples. We did this by completing a fixed effects and random effects meta-analysis using PLINK software, with results show in Author response table 1. Statistics shown are the N (sample size by group included in the meta-analysis), Fixed effects (p-value and effect size), random effects (p-value and effect size), Cochrane’s Q statistic as an indicator of variance across sample effect sizes, and the I heterogeneity index, which quantifies dispersion across samples. For some SNPs, the minor allele frequency in the African ancestry sample was below 1% and thus the subsample was excluded from the meta-analysis. The Q and I statistics indicate random effects analysis fit the data better for IADL and gait speed decline, thus we focus on results from random effects to account for differences in effect sizes by sample (e.g., the I index for IADL decline indicates 63.39% of the observed variance between samples is due to differences in effect sizes between samples). Given these results, the common effect size calculated for walking across a room and grip strength decline still suggest significance of these associations with SNPs in the gene, whereas the effects for IADL and gait speed decline remain for the European ancestry cohort only and not across subsamples. Further probing of the associations between SNPs within the gene and such phenotypes require physiological measures to target specific muscle groups.

**Author response table 1. sa2table1:** 

phenotype	SNPname	location	minor allele	European AncestryN	African AncestryN	HispanicN	Fixed EffectP-value	Random EffectP-value	Fixed Effect:OR or β	Random Effect:OR or β	Q	I
Walking across a room	Rs 111289603	19195492	G	9907	--	1067	0.00050	0.00050	1.4664	1.4664	0.5825	0.00
Grip strength decline	Rs 28665699	19200185	A	5228	--	409	0.00150	0.00150	-0.0418	-0.0418	0.3341	0.00
IADL decline (3 tasks)	rs111289603	19195492	G	9041	--	935	0.00316	0.07460	0.0644	0.0642	0.0984	63.39
Gait speed decline	rs77608580	19196968	A	3319	381	237	0.00775	0.72900	0.0424	0.0146	0.0577	64.95

2) Please link SNPs to ALDH4 expression or stability to provide some sense of how a SNP could result in functional changes--making this connection compelling underlies the major conclusion of the paper.

We emphasize that we have not identified a single variant that is responsible for driving the specific aging-related functional phenotypes tested in this particular study, rather we establish that there is variation within the gene that strongly suggests relationships with a common underlying factor. Further, it is likely that small effects of multiple SNPs across multiple genes, including within this same gene, and with non-additive effects (e.g., gene-by-environment effects), contribute to the resulting phenotypes. Without identifying a causal SNP, we can only aggregate available data to suggest what contributes to a biological pathway. For example, through exploitation of the publicly available Genotype-Tissue Expression (GTEx; https://pubmed.ncbi.nlm.nih.gov/29022597/) database, we found that a SNP within the gene was significantly associated with tissue-specific differential ALDH4A1 expression levels in whole blood, as shown in Figure 3—figure supplement 1. Further experimental studies to know the downstream effect of this altered gene expression or specific muscle functionality phenotyping, would be required to address the mechanisms.

We suggest that you mine the eQTL and RNA stability databases to determine which SNPs can be linked to some potential change in ALDH41A expression/stability. The eQTL analysis is both important to substantiate the claims and doable. What the authors would need to do to substantiate their claim that ALDH41A is a biomarker of muscle health in humans is to mine and perform the appropriate genetics and statistical analyses using existing cell-specific data in the extensive human eQTLs repositories (e.g., GTEx, eQTLGen, etc). Tissue-specific human eQTL data are publicly available, and can be extracted and analyzed. For more information on these repositories, the authors can refer to PMID: 34493866.

Invoking the Genotype-Tissue Expression (GTEx) database (https://gtexportal.org/home/), which is currently the most comprehensive resource for tissue-specific gene expression and regulation data (similar to the eQTL Catalog in the PMID mentioned), we find that there is an eQTL found in whole blood associated with one SNP in ALDH41A (rs77608580). This association with the eQTL implies there is some regulatory function related to the SNP, although further evaluation of muscle-specific effects using experimental approaches would be next steps for this research.

Better connect predictor human SNPs and *C. elegans* muscle aging biology.The SNPs of human ALDH4A1 were not analyzed with respect to the *C. elegans* alh-6 mutations. Reviewers agreed that making a "functional" connection from nematode to human SNP would strengthen the arguments put forward in the paper. Minimally, the authors should include clear listing of whether SNPs might be associated with a change in the *C. elegans* protein/transcript.

We have updated the text of the manuscript (thoroughly discussed above)

Easy additions for clarity and value:The goal of establishing a functional link between identified SNPs and ALDH4A1 function in human muscle can and should be enhanced by adding details regarding SNP impact on protein structure and function prediction, or of potential mRNA consequence (splicing site perturbation). Compare conserved amino acid sequences of *C. elegans* ALH-6 and human ALDH4A1 in parallel. Include schematic illustration of ALH-6 protein and predicted structures of ALH-6 wild-type and mutant proteins using protein structure prediction tools (i.e. AlphaFold2), which will provide useful data for alh-6 substitutions identified.

As suggested, we have used available computational resources to create models of the impact of each missense mutation on the protein structure of ALH-6 (Figure 1, figure supplement 2). We accomplished this by first modeling the wild-type ALH protein in Phyre2 – http://www.sbg.bio.ic.ac.uk/phyre2/html/page.cgi?id=index^2^. We then systematically modeled each missense mutation using http://missense3d.bc.ic.ac.uk/missense3d/^3^, which reports the impact, if any, to the protein structure. This data is summarized in Figure 1, figure supplement 3.

Suggested but not essential:Constructing human cognate CRISPR alleles in nematodes would be welcome (especially if any support a clear structure/function hypothesis), but since 1) it is possible that a SNP change might not directly correspond to nematode impact even if the proposed relationship were operative, and 2) such engineering could require a fair amount of elegans manipulation, making this connection is not considered a requirement for a successful revision.

In the future, we are certainly interested in testing structure/function hypotheses based on causal SNPs that we identify.

3) Comment on novelty of approach. The GWAS mining approach is important as it reports a success that opens up a novel avenue for connecting *C. elegans* biology/genetics to human physiology. Still, success with this approach for human obesity genes have been recently reported (PLoS Genetics; PMID: 34492009), so the approach is not the first of its type. Authors should cite that literature; their contribution on this front is still significant and the application in this system moves this approach toward a more central activity in the field, which is powerful.

This is an important point, and we now provide additional references of the literature for GWAS studies and testing in model systems. However, the novelty of our approach is the use of the large human cohort in the US Health and Retirement study that represents “normal aging” in the population and the use of the gene-wide association scanning (GeneWAS) approach. This is different than the more commonly applied genome-wide association studies (GWAS) that select for individuals with a specific disease state that drive morbidity and mortality.

4) The title should be precise and true to findings. R1 noted that "The title of this paper is somewhat misleading. The authors produced a prediction model of age-dependent decline of muscle functions based on genotypes of human ALDH4A1, but not those of *C. elegans* alh-6". The title. e.g., "Genetic variation in ALDH4A1 is associated with muscle health over the lifespan and across species" should be revised for accuracy.

We have changed the title as suggested

5) Explain better, write carefully on SKN-1 activity claims. R2 notes that gst-4::GFP can be induced by SKN-1 as well as by additional transcription factors; SKN-1 activity is not directly tested in this study. Authors should directly test SKN-1 activation, or just state gst-4::GFP expression as the outcome assayed, with caveats mentioned. The authors have previously produced data on skn-1 in this response, possibly more extensive discussion of that data might allay some of the concerns.

Although, we have previously demonstrated that the activation of the gst-4p::gfp reporter is dependent on SKN-1, we have adjusted the text as requested to indicate GFP expression from the gst-4p promoter as the assay outcome.

Authors should also consider R1 points about muscle-specific activation of gst-4::gfp claims.

This is an important point. Although the musculature is the most obvious response, our previous published studies reveal the expression was restricted to body wall muscle, pharyngeal muscle and neurons. We have included a comment about neurons in our manuscript.

In sum, in revision, the authors precise as to what is directly assayed in the screen (gst-4::GFP expression, not necessarily SKN-1 activity) in the summary.

See point 9 above.

Reviewer #1 (Recommendations for the authors):1. I think the biggest issue for this excellent paper is that the information about SNPs of human ALDH4A1 was not analyzed with respect to the *C. elegans* alh-6 mutations. Do the SNPs occur at the same or similar loci of the orthologous *C. elegans* mutation sites? If not, can the authors introduce human SNPs into C. elegans for the orthologous changes and test whether they affect age-dependent declines in muscle functions? This is the key for improving the paper to fit the purpose of the work.

We thank the reviewer for the positive assessment for our work. Due to limitations in the assessment of the genomic data from the large human cohort, which is not whole genome sequencing but an Illumina array of 2.4 million SNPs, so we are unable to assess complete variation in Aldh4a1. It is possible to use imputed SNPs, but these would not augment our ability to identify additional or causal variants within the gene because the imputed ones are already represented by marker SNPs in high linkage disequilibrium that are present on the genotyping array. Thus, the available SNPs are those that best represent common variation within the gene given the current array technology. However, the association of multiple SNPs in Aldh4a1 with age-associated loss of muscle health is highly suggestive that variation at this locus (perhaps attributable to a SNP not directly measured by the array) is linked to function. Once fully sequenced it would be of great interest to test any SNPs that change protein coding regions and compare with the frequency of mutations in those homologous regions in ALH-6 recovered from our genetic screens.

2. The title of this paper is somewhat misleading. The authors produced a prediction model of age-dependent decline of muscle functions based on genotypes of human ALDH4A1, but not those of C. elegans alh-6. Please downplay the title. e.g., "Genetic variation in ALDH4A1 is associated with muscle health over the lifespan and across species"

We have changed the title as suggested. “Genetic variation in ALDH4A1 is associated with muscle health over the lifespan and across species”

3. In Figure 1b, please show schematic illustration of ALH-6 protein. In addition, predicted structures of ALH-6 wild-type and mutant proteins using protein structure prediction tools (i.e. AlphaFold2) will provide useful information for alh-6 mutations (for example, "G152/K153" in page 5 line 89).

Figure 1b is indeed a schematic illustration of the ALH-6 protein. We have modified the legend to explicitly state this.

4. In Figure 1c, please describe accurately how the authors executed One-way ANOVA statistical analysis for measuring GFP intensity.

We apologize for the confusion, we initially performed a one-way ANOVA to compare the gst-4p::gfp reporter strain, the alh-6(lax105) canonical allele, and the new alleles isolated from our screen. However, to make the analysis more clear we have modified our statistical analysis to evaluate each mutant relative to the gst-4p::gfp control.

5. I recommend comparing conserved amino acid sequences of *C. elegans* ALH-6 and human ALDH4A1 in parallel.

We have added the % identity for ALH-6 and ALDH4A1 in the introduction.

6. In figure supplement 1, please add a scale bar.

We have added a scale bar.

7. In Figure 2, please elaborate the choice of p values 0.003 and 0.013.

We agree this approach of p-value determination is important to carefully address. With agnostic genome-wide association studies, the gold standard method is to adjust the p-values based on stringent thresholds, most simply and conservatively done through Bonferroni-corrected p-values (calculated as 0.05/number of SNPs tested*independent traits tested). This is because it is unknown whether any variation in the gene is associated with the traits of interest. However, this investigation was not agnostic, rather there was a hypothesized association between variation in the gene and age-related muscle function from prior experimental results as described in *C. elegans*. In subsequent association studies, in order to validate a hypothesized association, a threshold of p<0.05 has been used; however, this raises the potential for type 1 error or reporting false positives. Thus, we clarify our approach more thoroughly below -- under “High priority-Essential” item #1.

We note that some SNPs represent rarer variants in humans depending on the population from which the sample was drawn. Also, it is unknown the degree to which the SNP frequency represents conservation of variation in a model organism like *C. elegans* because frequencies are highly dependent upon human ancestry. Thus, we chose to include results for all SNPs with a minor allele frequency of at least 0.01 as each of these SNPs serve as markers for other SNPs within the gene that may not be represented on the array.

8. In Figure 3, please mark exact genotypes at x axes.

We have updated the figure as requested. The counts represent the number of minor alleles for each SNP (which is among European ancestry individuals):

**Author response table 2. sa2table2:** 

SNP name	0	1	2
Rs77608580	GG	AG	AA
Rs28665699	GG	AG	AA
Rs111289603	AA	AG	GG

9. For Figure 3, p values are in written the Figure Legends and Table 1. However, it will be better to mark p values or asterisks in the panels as well.

We have left the figures as is and emphasized in the text that these effects are examples of where there is variation in the gene contributing to phenotypes that represent a range of age-related change in functionality; overall we find there are small effects associated with each phenotype, there are possible pleiotropic effects, and the SNPs identified represent markers for variation within the gene, but we cannot claim causality.

10. I recommend adding discussions regarding how genetic variants in alh-6/ALDH4A1 contribute to age-dependent impairment in muscle functions. In particular, it will be great to add speculation for changes in mitochondrial proline dehydrogenase levels and/or activities that may affect sarcopenia.

See response under the “High priority-Essential” item #1.

The traits used in these scans came from a large and sociodemographically diverse, naturally aging human sample so that they do not allow us to identify physiological degeneration or impairment in specific muscles, rather to index and track overall decline in functionality through older ages. It is widely accepted now that the genetic variation underlying these aging-related traits are highly polygenic. Thus, it is not expected that a single variant within a gene would be identified to drive these phenotypic changes in humans. It is likely that small effects of multiple SNPs across multiple genes, including within the same gene, and with non-additive effects (e.g., gene-by-environment effects), contribute to the resulting phenotype, or multiple phenotypes (i.e., pleiotropy). With this use of genewide-association scanning approach, it is only possible to identify those variants associated with overall effects.

11. Please change one of "IADLA Decline" to "IADLA2 Decline" in Figure 2.

We have changed the text and modified the figure

12. On page 5, line 85, add lax918 in addition to lax903 and lax930 for E78Stop.

We have changed the text

13. On page 6, line 163, use "day 3" instead of "Day 3" for consistency.

We have changed the text

Reviewer #2 (Recommendations for the authors):To strengthen the main claims this reviewer suggests a revision of the following points:1) The authors state:"Based on the striking specificity of the muscle-restricted and age-dependent activation of SKN-1 in *C. elegans* harboring mutations in alh-6,…"However, currently, data showing SKN-1 activation in the alh-6 mutant backgrounds are lacking. Direct observation of SKN-1::GFP activation would be important because the transcriptional reporter gst-4P::GFP, although a target of SKN-1, is also activated by other stress-responsive TFs including NHR-49 (PMID: 30297383) and DAF-16 (PMID: 32161088).In addition, in Suppl. Figure 1 various alh-6 mutant alleles show non-muscle gst-4P::GFP signal.Therefore, the authors may need to directly test SKN-1 activation, and revise the muscle-specific claims.

We thank the reviewer for this suggestion. Although we have previously shown that the activation of the gst-4p::gfp reporter is lost from skn-1 RNAi treatment, we have edited the text to clarify and specify the observation of the muscle specific activation of the gst-4p::gfp reporter.

2) Structural predictions may enable the authors to generate hypotheses suggesting how a very early STOP codon in alh-6 (e.g. lax906) may have a phenotypic impact on muscle function comparable to a late amino acid substitution (e.g. lax947).

This is indeed an interesting question for future structure-function studies. We have used Phyre to model predicted protein structure of wild-type ALH-6, and used Missense3D to locate the mutations on the predicted protein structure. See response under “High priority-Essential” item #5 below.

Previous studies have demonstrated that the level of mitochondrial dysfunction (and ROS production) is related to both beneficial and detrimental physiological outcomes; even driving lifespan extension versus lifespan shortening (reviewed in PMID:33644065).

3) The authors tested the significance of 53 SNPs in the ALDH4A1 locus with 16 phenotypes in the HRS (Table S1). They claim that they are performing a gene-wide association study and are correcting for 16 tests (0.05/16) in their statistical significance assessment. This analysis is not correct. The authors need to correct for 53x16=848 tests (0.05/848 = 5.8e-5 if they choose to use Bonferroni correction). The authors can argue that the SNPs are not independent, which would be true. In that case, they need to prune the SNPs based on linkage disequilibrium and use index/tag SNPs for their analyses. Further, some of the 16 phenotypes could also be highly correlated; and this needs to be acknowledged/addressed.

See response under the “High priority-Essential” item #1.

4) It is unclear how the SNPs presented in Table 1 are linked to ALDH4A1. The SNPs are in the locus, but that does not necessarily mean they have a functional impact on ALDH4A1. For instance, do the identified SNPs lead to amino acid, splicing, or other regulatory changes in ALDH4A1?Ultimately, this paper aims to advance ALDH4A1 as genetic factor underlying muscle-function decline in aging humans. Therefore, to go beyond the already publicly available link between the SNPs in the ALDH4A1 locus and muscle performance in aging humans, the authors may need to define whether the reported SNPs are linked to ALDH4A1 expression as eQTLs, especially in muscle tissue/cells.

This is an important point, which we have clarified further in the text. Due to limitations in the assessment of the genomic data from the large human cohort, which is not whole genome sequencing but an Illumina array of 2.4 million SNPs, so we are unable to assess complete variation in Aldh4a1. It is possible to use imputed SNPs, but these would not identify additional or causal variants within the gene because the imputed ones are already represented by marker SNPs in high linkage disequilibrium that are present on the genotyping array. Thus, the available SNPs are those that best represent common variation within the gene given the current array technology. However, the association of multiple SNPs in Aldh4a1 with age-associated loss of muscle health is highly suggestive that variation at this locus (perhaps attributable to a SNP not directly measured by the array) is linked to function. Once fully sequenced it would be of great interest to test any SNPs that change protein coding regions and compare with the frequency of mutations in those homologous regions in ALH-6 recovered from our genetic screens.

5) For clarity, the distribution of the phenotypes for each allele may be presented in Figure 3 like they are presented in Figure 4.

The distribution of phenotypes for each variant is shown in Figure 2. We have updated figure 3 to show the effect of each allele.

Reviewer #3 (Recommendations for the authors):1) The finding that alh-6 mutants exhibit declines in mobility during aging could reflect a selective effect on muscle function, or could be reflective of a larger acceleration of aging. It would be helpful to show lifespan data for the alh-6 mutant, or discuss if this work has been published previously.

We have referenced the previous work of Pang et al., Which documents the lifespan data of the original alh-6 mutant alleles.

2) The conclusion that the alh-6 mutant affects muscle function during aging could be further bolstered via studying muscle mass and structure as well as mitochondrial number and structure will readily available reporters.

We have referenced the previous work of Pang et al., and Yen et al., which documents the mitochondrial measures of alh-6 mutants.

3) For the human SNP analysis, engaging the help of an expert in human Kinesiology or Geriatrics can help with narrowing the phenotypes to make them more selective for muscle function, and exclude those where the connection to muscle function is more tangential.

We appreciate the reviewer’s suggestion. We have chosen to be more inclusive of the data available in the HRS, which is a population-based sample with rich phenotyping across multiple traits. These traits were not assessed to allow us to identify physiological degeneration in specific muscles, rather to index and track overall age-related decline in functionality through a large and sociodemographically diverse, naturally aging human sample. Thus, the inclusion of multiple indicators allows us to provide some degree of sensitivity testing using the population-based collection of measures, and thus greater confidence that normative human muscle functioning and change in functioning over age finds some association with variation in this gene. We have now included in the text the notion that some HRS phenotypes will be more directly related to muscle function while others are more complex and are thus more likely to depend on additional factors.

4) Several recent papers have used genome-wide association analysis to identify SNPs associated with decreased muscle strength in middle-aged and older individuals. This work will provide information on other readily available dataset that could be suitable for a replication study to determine if the findings are seen in separate study populations.

We agree and we have referenced these data sets, although they require application, review and approval to use the data for new studies. Any of the HRS family of sister studies that are collecting genetic data: English Longitudinal Study of Ageing (ELSA; https://www.elsa-project.ac.uk); Irish Longitudinal Study on Ageing (TILDA; https://tilda.tcd.ie/); cohorts in the Survey of Health, Ageing and Retirement in Europe (SHARE; https://g2aging.org/overviews?study=share-aut), or Northern Ireland Cohort for the Longitudinal Study of Ageing (NICOLA; https://www.qub.ac.uk/sites/NICOLA/AboutNICOLA/); and others that include European ancestry individuals who are aged 50 and older.

5) For the SNP studies, the 0.003 p-value cut-off is likely too low given that 53 SNPs are being studied making the multiple testing greater than the correction used.

We agree that with agnostic genetic association testing, designed for the discovery of whether there is an association between variation within a gene and a trait, that a cut-off with the 53 SNPs would be more appropriate. However, in this case, the genetic association testing was designed to test an a priori hypothesis given findings presented in *C. elegans*. We chose the 0.003 cut-off for several reasons, primarily that (a) this study was designed to validate an existing hypothesis; (b) that each of the 53 SNPs within the gene represents a tag, or marker SNP for human variation within the same gene, such that with this association testing we are not be able to identify a causal variant, rather only to index that some variation within the gene suggests conserved associations with humans; and (c) because it is unknown which SNPs represent conservation across species, particularly among humans, for whom there is genetic variation due to ancestral histories, we did not want to miss potential SNP associations that do represent important variation within the gene. We have acknowledged this rationale in the text as well as the limitations of these methods for providing us with a complete understanding of the molecular genetic and etiological basis for age-related muscle-functioning and decline.

References cited in response to reviews:

1. Battle, A. et al. Genetic effects on gene expression across human tissues. Nature 550, 204-213, doi:10.1038/nature24277 (2017).

2. Kelley, L. A., Mezulis, S., Yates, C. M., Wass, M. N. and Sternberg, M. J. The Phyre2 web portal for protein modeling, prediction and analysis. Nat Protoc 10, 845-858, doi:10.1038/nprot.2015.053 (2015).

3. Ittisoponpisan, S. et al. Can Predicted Protein 3D Structures Provide Reliable Insights into whether Missense Variants Are Disease Associated? J Mol Biol 431, 2197-2212, doi:10.1016/j.jmb.2019.04.009 (2019).

[Editors’ note: further revisions were suggested prior to acceptance, as described below.]

Thank you for resubmitting your work entitled "Genetic variation in ALDH4A1 is associated with muscle health over the lifespan and across species" for further consideration by eLife. Your revised article has been evaluated by Carlos Isales (Senior Editor), a Guest Reviewing Editor, the original reviewers, and an expert in statistics. We are sorry about the delay.While Reviewer 1 is now in favor of acceptance as is, Reviewer 3 states: "I still have concerns about the multiple testing correction being used by the authors, and I did not feel that the authors addressed this in a fully compelling manner, such as adding a statistician."

We have implemented suggestions from Reviewer 3 (described in further detail below) and focus on two specific two phenotypes (grip strength and gait speed) instead of all sixteen as previously analyzed. Although this reduces the depth of the data presented, we have accommodated the request of the reviewer. As requested, we have consulted with Wendy Mack, PhD, who is a professor of biostatistics in the Department of Preventive Medicine at the Keck School of Medicine of USC. She co-directs the department’s Division of Biostatistics graduate programs and directs Biostatistics Resources at the Southern California Clinical and Translational Science Institute (SC CTSI).

Reviewer 2 concurs: "Although I would really like for the work to be complete, I'd need to agree with Reviewer 3. In addition to unjustified processing of the data that leads to a significance level (0.05) that seems inappropriate for this type of study, the authors have not addressed the concerns related to experimental testing of some of the key claims in *C. elegans*. For instance, the authors did not directly look at SKN-1::GFP even though the tools are available and this is a feasible experiment. As explained before, the current readout is indirect (gst-4::GFP), and although gst-4 is a target of SKN-1, it is also a target of DAF-16 and other transcription factors. Therefore, gst-4::GFP levels are only suggestive of SKN-1 activation. I agree that adding a GWAS expert statistician would be really helpful."

We are not sure what this reviewer is asking; with the exception of the introduction section’s reference to past literature about stress pathways, we do not mention SKN-1 (or DAF-16) anywhere in our results or discussion in the manuscript. We have removed the two instances of usage of the word "SKN-1" from the introduction as this does not change our manuscript and the references provided make this point for us.

eLife consultant on statistics questions added: "It is true that in candidate gene association you may not need to correct for multiple tests, however, this is only the case when SNPs used are known to have a function (usually non-synonymous coding). I support the suggestion made by reviewers to use linkage disequilibrium (LD) for SNP pruning and also reduce the number of phenotypes tested if possible.

We have reanalyzed all the data and focus the results on the SNPs that are most representative after pruning and for targeted phenotypes as described below.

Regarding SNPs, the authors' suggestion that all 53 SNPs are "tags" of unknown functional variants is not justified. How were they selected? If they are correlated, many of them still do tag the same function. Whether pruned or not, SNP significant association from this analysis is only an association, not causality especially given the complex nature of the phenotypes. It is true that LD pruning would make it difficult to replicate findings in other studies, particularly from other populations but it would be possible to use the nearest SNP available in those studies (as would also be the case if a different genotyping array was used).

We now see where our language was confusing and edited the sentence (top of page 5) for clarity as well as edit the description in the methods in the last two sentences under Genotyping Data. SNPs selected for inclusion involved initially using all present on the Illumina Omni2.4 array, which totaled 70. After filtering for minor allele frequency of 0.01 or greater, 53 SNPs remained.

We also completely agree that the SNPs included are designed to be tags and results can represent only associations, without any implications for causality as explained in the revised manuscript. We have revised sentences that were confusing in this respect including removing the last sentence at the end of first paragraph on page 5, editing the last sentence of the introduction (removed the word “predictive” with “associates with”), and summary (revised wording from “impact” to “associations”).

The reviewers have provided helpful suggestions and in order to identify the most representative SNPs for potential functionality and efficiency in approach, we pruned SNPs and re-ran the GeneWAS with the refocus on fewer primary phenotypes. In the Methods, we now describe the process of filtering for minor allele frequency and pruning based on LD. This yielded 21 SNPs for consideration in the GeneWAS. We revise the corresponding methods and results to reflect this.

Regarding phenotypes, first of all it is good to have such a richly annotated data. However, as already acknowledged, these are not independent and correcting for 16 tests is unduly self-penalising (just as it would be to correct for 53 SNPs). It would be more appropriate to set one phenotype (or a few unrelated) as the primary focus and others as exploratory or explanatory of the main ones if a link can be established between them (which I think there is)."

We agree with the correlated nature of the phenotypes and appreciate the reviewer’s emphasis on the self-penalizing nature of the study design. In order to strengthen the analytical approach and focus on phenotypes that better address the goal of testing associations between genetic variation in ALDH4A1 and normative aging-related muscle functioning (vs. endurance, cardiovascular fitness, or stamina), as reviewer suggests, we have prioritized the primary focus onto main phenotypes that are more robust indicators of functionality and more precise measures of aging-related change (grip strength decline, gait speed decline) such that the two index change using repeated measures of the phenotype data rather than single time-point assessments. Instead of employing a Bonferroni adjustment for multiple-test correction, we now invoke permutation testing to set an empirical p-value threshold with which to evaluate the significance of associations. The presentation of methods and results has been revised based on this approach.